## REPORT

# A direct role for SNX9 in the biogenesis of filopodia

Iris K. Jarsch[1]*, Jonathan R. Gadsby[1]*, Annalisa Nuccitelli[2], Julia Mason[1], Hanae Shimo[1], Ludovic Pilloux[3], Bishara Marzook[1], Claire M. Mulvey[1], Ulrich Dobramysl[1], Charles R. Bradshaw[1], Kathryn S. Lilley[4], Richard D. Hayward[3], Tristan J. Vaughan[2], Claire L. Dobson[2], and Jennifer L. Gallop[1]

**Filopodia are finger-like actin-rich protrusions that extend from the cell surface and are important for cell–cell communication and pathogen internalization. The small size and transient nature of filopodia combined with shared usage of actin regulators within cells confounds attempts to identify filopodial proteins. Here, we used phage display phenotypic screening to isolate antibodies that alter the actin morphology of filopodia-like structures (FLS) in vitro. We found that all of the antibodies that cause shorter FLS interact with SNX9, an actin regulator that binds phosphoinositides during endocytosis and at invadopodia. In cells, we discover SNX9 at specialized filopodia in _Xenopus_ development and that SNX9 is an endogenous component of filopodia that are hijacked by _Chlamydia_ entry. We show the use of antibody technology to identify proteins used in filopodia-like structures, and a role for SNX9 in filopodia.**

## Introduction

Assemblies of actin filaments (F-actin) are major dynamic superstructures required for cell motility, intercell communication, and force generation. Filopodia contain parallel, tightly bundled arrays of F-actin and protrude from the cell to traverse substantial distances to detect substrate stiffness, the extracellular matrix, and initiate cellular responses to growth factors. Filopodia are hijacked by bacterial and viral pathogens during cell entry and can also facilitate cell-to-cell transmission (Chang et al., 2016).

Despite the fundamental importance of filopodia, as yet there is no unifying model to explain how the actin cytoskeleton and membrane machinery initiate, elongate, and sustain the dynamic behavior of filopodia. While filopodia appear to arise from Arp2/3-dependent actin polymerization (Yang and Svitkina, 2011), there may also be other mechanisms of filopodia initiation, such as via formins (Faix et al., 2009). Ena/VASP proteins are important in filopodia formation in cortical neurons and retinal ganglion cells, where their roles in elongating F-actin are thought to be direct (Dent et al., 2007; Dwivedy et al., 2007). However, in osteosarcoma cells, although reduced levels of Ena/VASP proteins inhibit filopodia, they localize to focal adhesions in the cell body rather than at filopodia tips, which suggests an indirect role (Young et al., 2018).

Cell-free approaches allow the functional reconstitution of specific actin assemblies and the biochemical analysis of their components in response to lipid signals. Dynamic filopodia-like structures (FLS) can be reconstituted using high-speed supernatant _Xenopus_ egg extracts and supported lipid bilayers augmented with phosphatidylinositol (4,5) bisphosphate (Lee et al., 2010). Our previous work has demonstrated that the initial stage

of FLS growth is sensitive to inhibition of Arp2/3 complex actin nucleation whereas steady-state growth dynamics involve an interchanging complement of F-actin elongation and bundling proteins including the formin Drf3 (mDia2), Ena, VASP, and fascin (Dobramysl et al., 2019 _Preprint_; Lee et al., 2010).

Here we have employed phage display phenotypic screening for the de novo identification of proteins involved in making FLS. In this approach, human single chain variable region fragments (scFv) are displayed on the surface of bacteriophage with the corresponding gene contained within their DNA. Phage of interest can be selected in a binding assay, the sequence of the variable region determined, and a specific antibody expressed and purified. The antibodies are then used in a functional assay to select those that generate phenotypes of interest, with the antigen identified by protein array or immunoprecipitation followed by mass spectrometry. While antibodies have much recognized success in perturbing function extracellularly as therapeutics and research tools, the use of cell-free systems enables access to intracellular antigens.

We have identified antibodies against FLS that induce a variety of morphological changes in the actin bundles. By selecting the antibodies that generated shorter FLS, we identified the antigen recognized by three independent antibodies as sorting nexin 9 (SNX9) and subsequently demonstrated its role by immunodepletion and rescue experiments. Using human cell culture and _Xenopus_ embryo explants, we have directly localized SNX9 to filopodia. Thus, we show that phage display phenotypic screening represents a powerful approach for identifying novel proteins in cell-free systems, and the involvement of SNX9 in filopodia.

[1]Gurdon Institute and Department of Biochemistry, University of Cambridge, Cambridge, UK;   [2]Antibody Discovery and Protein Engineering, AstraZeneca, Granta Park, Cambridge, UK;   [3]Department of Pathology, University of Cambridge, Cambridge, UK;   [4]Department of Biochemistry, University of Cambridge, Cambridge, UK.

*Iris K. Jarsch and Jonathan R. Gadsby contributed equally to this paper;   Correspondence to Jennifer L. Gallop: j.gallop@gurdon.cam.ac.uk.

# Results and discussion

## Antibody-mediated modifications of FLS by phage display phenotypic screening

To isolate antibodies targeting novel components of FLS, a phage display library was incubated with a cocktail of purified actin regulatory proteins that are known to localize to FLS to adsorb and deselect phage against known FLS components (actin, TOCA-1, Ena, VASP, N-WASP, fascin, and the Arp2/3 complex; Lee et al., 2010). A further deselection step was performed against the supported lipid bilayer and experimental setup, followed by positive selection of phage on FLS (Fig. 1 A). We performed screens against mature fully formed FLS, mature FLS subjected to an additional washing step to relax the bundled FLS architecture, and at an early time point where FLS were fixed rapidly to ensure accessibility to the earliest arriving proteins. Phages selected from each condition were eluted, cloned, and sequenced. We excluded phages that bound under all three conditions as they were more likely to bind residual proteins from the extracts. Phage ELISA screening of each selection output was performed against their respective condition and the bilayer alone, and a panel of highly specific, strongly binding clones was selected from each condition (Fig. 1 B). A single phage that bound to the cocktail of known proteins was also selected (scFv8). We purified 22 scFvs, which were individually preincubated with the assay mix and resulting FLS visualized by spinning disk confocal microscopy. We performed the screen with and without additional unlabeled actin as lengthening phenotypes could be constrained by limitations in the actin, whereas shortening phenotypes could be suppressed by greater actin concentrations.

Remarkably, a range of altered FLS phenotypes was evident upon coincubation of different scFvs (Fig. 1 C). We quantified FLS parameters using our image analysis pipeline FLSAce and used the resulting median number, lengths and areas both with and without additional actin as the input for the t-distributed stochastic neighbor embedding (tSNE) algorithm to visualize similarities between the actions of scFvs (Fig. S1 and Fig. 1 D). The tSNE plot indicates several clusters (Fig. 1 D): scFvs 1, 9, and 19 are high in numbers, slightly longer and less straight; FLS in scFv2 and scFv11 are among the longest and thinnest (i.e., have a small FLS base area); in contrast, scFvs 6, 7, 8, and 14 are long as well, but thicker; scFv10 and scFv17 are moderately high in numbers with long FLS; scFvs 3, 4, 5, 18, and 20 show slightly higher numbers and a reduction in FLS length; scFvs 12, 13, and 15 are slightly longer and show slightly higher numbers; and scFvs 16, 21, and 22 show varying degrees of similarity to the control.

The antibodies could be either blocking a single protein with a single activity, giving a distinct phenotype; blocking a single protein, giving complex effects due to the multiple roles of the target protein; or reacting with more than one sequence related protein and/or against a protein complex giving either simple or complex phenotypes. Since the shorter phenotype could correspond to the inhibition or sequestration of factors involved in FLS formation or elongation, we focused our effort on the further characterization scFv3 and scFv4 that generated substantially shorter FLS.

## SNX9 as a key mediator of FLS formation

To identify the target antigens, we probed Xenopus egg extracts by immunoblotting with scFvs 3 or 4, with both recognizing a 70-kD species (Fig. 2 A). To enable immunoprecipitation, the two scFvs were converted to an IgG1 framework to generate monoclonal antibodies with higher avidity (correspondingly, IgFls3 and IgFls4). These antibodies also recognized a 70-kD species in Xenopus egg extracts (Fig. 2 B). Following separation by SDS-PAGE and size exclusion by excision of regions of the gel spanning ~50–80 kD, eluted proteins were identified by mass spectrometry by comparing extracts, immunoprecipitation (IP) with control IgG and IP with IgFls3 (Fig. S2, A and B). The highest scoring target of the right molecular weight was SNX9 (Fig. S2 C), a phox homology bin-amphiphysin-rvs (PX-BAR) protein involved in endocytosis and control of actin polymerization (Yarar et al., 2007). Indeed, purified SNAP-tagged SNX9 was recognized by both scFv3 and scFv4 by immunoblotting (Fig. S2 D).

To validate the requirement for SNX9, we used a previously raised rabbit polyclonal antibody (Gallop et al., 2013) to immunodeplete SNX9 from Xenopus egg extracts and blotted with IgFls3 and IgFls4. This showed a complete loss of the previously recognized species at ~70 kD (Fig. 2 B). A comparable result was obtained when identical samples were immunoblotted with the anti-SNX9 antibody, definitively identifying the ~70-kD species as SNX9 (Fig. 2 B). FLS generated using the SNX9-depleted Xenopus egg extracts were significantly shorter (Fig. 2 C), a phenotype rescued by the addition of purified SNX9 (Fig. 2 C). The immunoblock induced by preincubation of extracts with scFvs 3 or 4 could also be overcome by the addition of purified SNX9 (Fig. 2, D–F; statistics are in Table S1). We additionally looked at scFvs 5 and 21, which gave similar shorter FLS phenotypes in the initial screen (Fig. 1, C and D; and Fig. S1). Both also recognized a 70-kD species in extracts; however, only scFv5 recognized purified SNX9 (Fig. S2 E). Similarly, while scFv5-dependent inhibition of FLS was effectively relieved following the addition of purified SNX9, this was less effective for scFv21 (Fig. 2, D–F; and Table S1), suggesting that scFv21 recognizes a related protein rather than SNX9 itself.

To establish directly whether SNX9 localizes to FLS, we supplemented the reaction mix with purified labeled SNAP-SNX9, seeing enrichment at the tips of FLS where the actin bundle grows from, marked by actin barbed end polymerase VASP (Fig. 2 G). Equivalent localization was observed when fixed FLS were immunostained with an anti-SNX9 antibody (Fig. S2 F). Thus, SNX9 localizes to the membrane-proximal tip, and is therefore likely a membrane-binding regulatory component rather than a structural element distributed throughout the actin bundle. Together, these data identify SNX9 as an important mediator of FLS formation.

SNX9 is known to activate actin polymerization by binding N-WASP downstream of its interaction with membranes, and thereby stimulate Arp2/3 complex activation (Daste et al., 2017; Gallop et al., 2013; Yarar et al., 2007). In cancer cells, overexpression of SNX9 is thought to cause filopodia formation by promoting Cdc42 activation, but the protein domain responsible is not known (Bendris and Schmid, 2017; Bendris et al., 2016a,b).

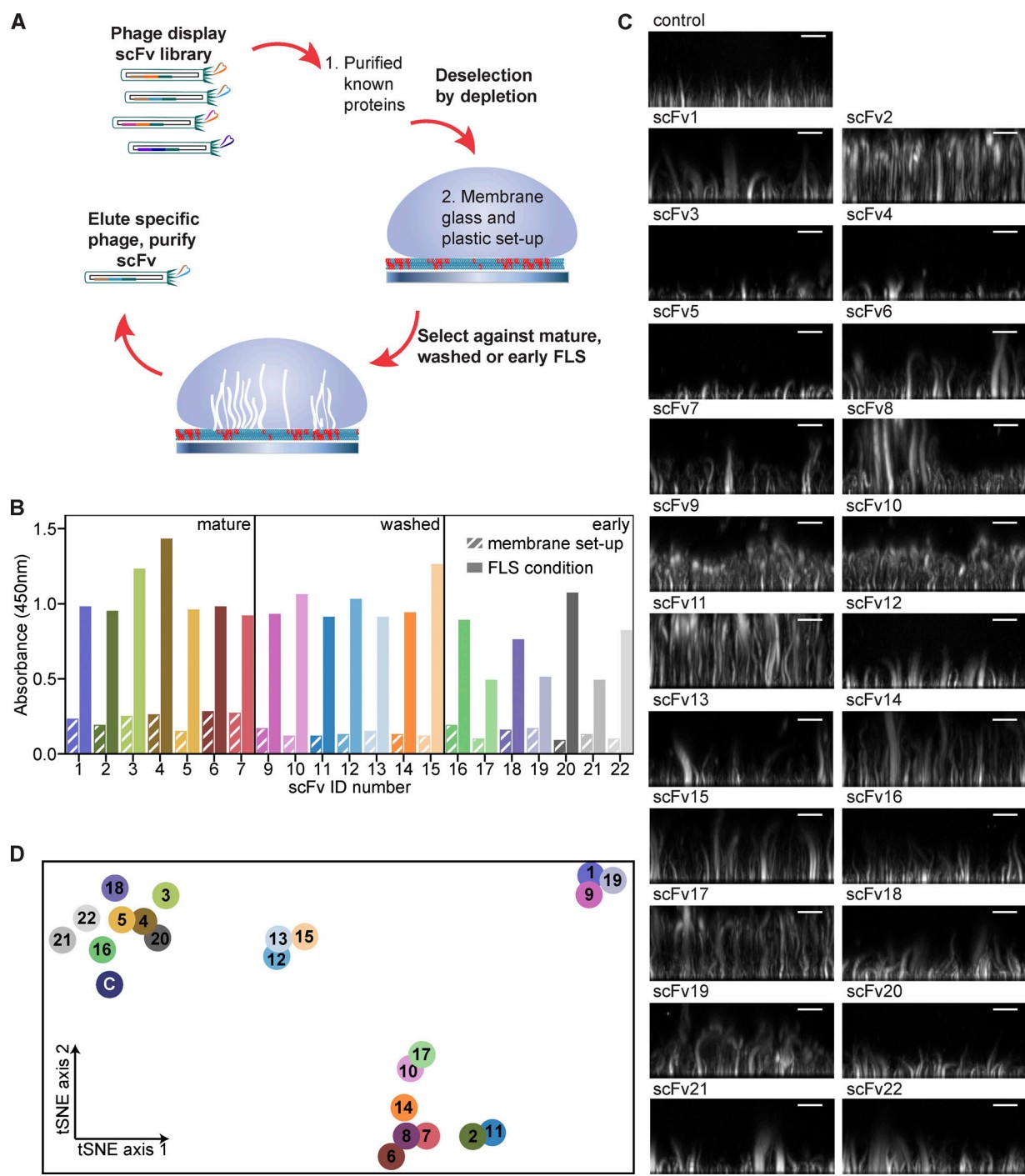

Figure 1. **Phage display phenotypic screen on FLS. (A)** Schematic of the screen. **(B)** Phage ELISA yes/no screening on selection outputs ($n$ = 1) leading to the identification of 22 unique phage clones specific to their FLS condition with lack of binding to the bilayer and setup (hatched boxes). **(C)** Example images of FLS phenotypes on preaddition of 5 µl of each scFv of ∼1 mg/ml concentration to the FLS assay mix. Images are maximum intensity projections of 1 µm confocal Z stack reconstructions viewed from the side. Scale bars, 10 µm. **(D)** T-distributed stochastic neighbor embedding plot of the 22 scFvs and buffer control ("C"). The distance between the points shows a 2D representation of the 6D neighborhood structure spanned by each condition's median FLS count, the median average FLS length, and the median average FLS base area, for both additional actin added and no actin added. Each scFv is color coded.

Using liposomes of defined lipid composition and *Xenopus* egg extracts, which produces smaller actin assemblies compared with FLS, we have previously found that the membrane interaction of SNX9 occurs via a dual interaction with phosphatidylinositol 4,5-bisphosphate ($PI(4,5)P_2$) via the BAR domain and PI(3)P with the PX domain (Daste et al., 2017; Gallop et al., 2013). Since phosphatidylinositol 3,4,5-trisphosphate ($PI(3,4,5)P_3$) and phosphatidylinositol 3,4-bisphosphate ($PI(3,4)P_2$) are implicated in filopodia formation via phosphoinositide 3-kinase (Jacquemet et al., 2019; Johnson and Haugh, 2016), we investigated the effect

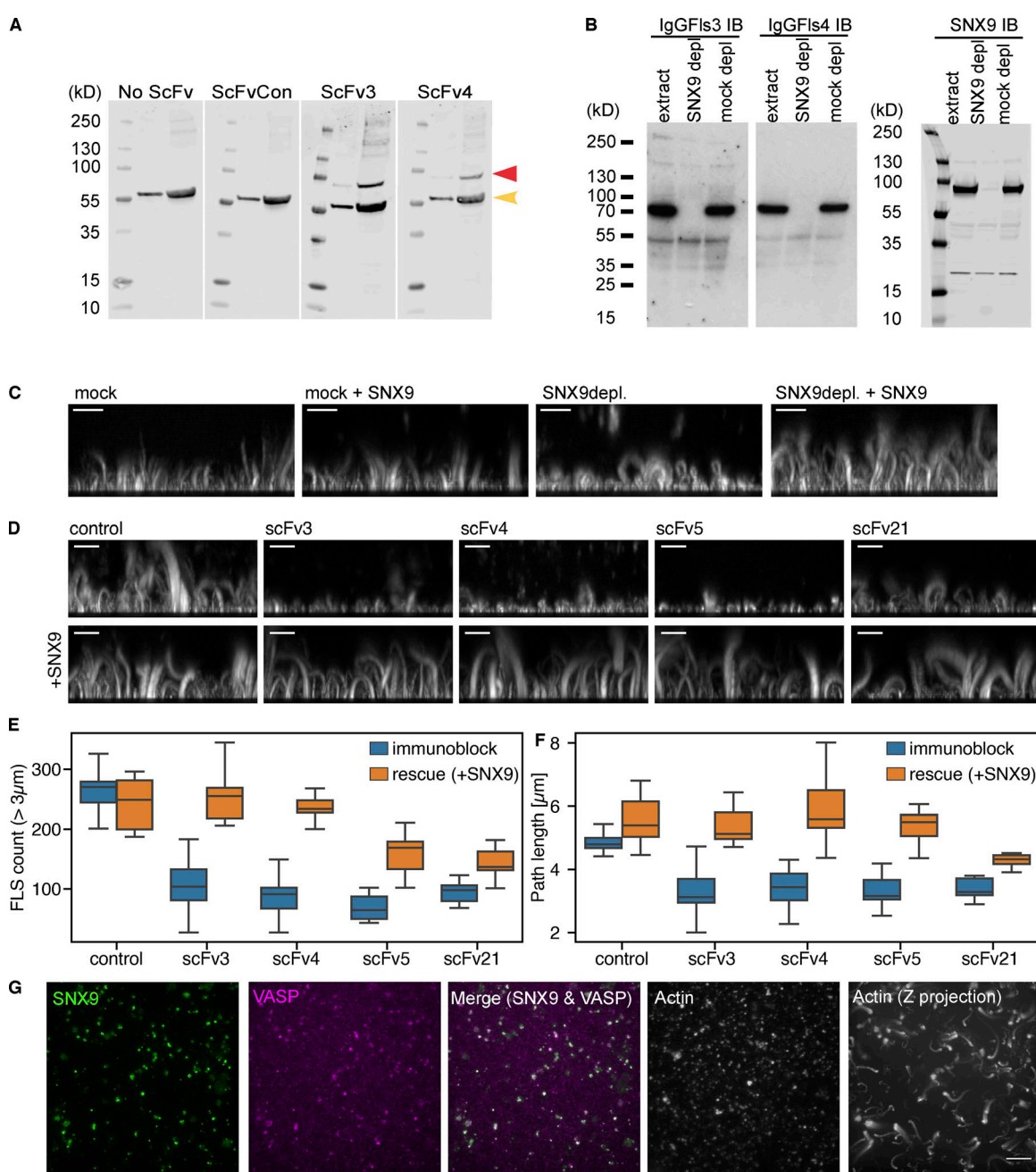

Figure 2. **Identification of SNX9 as the antigen to antibodies giving a shorter FLS phenotype. (A)** ScFv3 and 4 both detect a specific band not detected in either control sample (red arrow) by Western blotting (left = 25 µg, right = 100 µg). The band at ∼60 kD in all four blots (yellow arrow) is nonspecific reactivity of the secondary or tertiary antibodies with extracts. **(B)** Western blots of control and SNX9 or mock-depleted extracts with IgGFls3, IgGFls4, or rabbit-anti SNX9. The ∼70 kD band is lost on SNX9 depletion. **(C)** Side maximum intensity projections of 1 µm confocal z-stacks of either mock or SNX9-depleted extracts. Immunodepletion with anti-SNX9 antibodies reduces FLS length and is rescued by 20 nM purified SNX9. **(D–F)** Immunoblock (IB) by specific scFvs and rescue by 20 nM purified SNX9. **(D)** Side maximum intensity projections of 1 µm confocal z-stacks of FLS grown using extracts immunoblocked with control or specific scFVs. Reductions in FLS length and number can be rescued by addition of purified SNX9. **(E and F)** Quantification shows the median, quartiles, and range, n = 9–12 imaging regions per condition from two independent experiments. Statistics are in Table S1. **(G)** TIRFM shows colocalization of SNX9 and VASP with actin and Z-maximal projection of a 1-µm confocal stack of actin in FLS. Scale bars, 10 µm.

of inhibiting phosphoinositide 3-kinase on FLS and asked which phosphoinositides were involved in FLS formation.

Recruitment of SNX9 and FLS formation was markedly reduced in the presence of phosphatidylinositol 3-kinase (PI3K) inhibitors LY294002 and wortmannin, while the recruitment of Fes/CIP4 homology-BAR (F-BAR) domain actin regulatory protein

TOCA-1 was unchanged (Fig. 3, A and B). Using purified phosphatidylinositol 3-phosphate (PI(3)P)-responsive mCh2xFYVE as a lipid reporter, we could show that PI(3)P is produced from supported lipid bilayers containing PI(4,5)$P_2$, where it colocalizes with sites of FLS growth (Fig. 3 B; statistics are in Table S2). We then applied wortmannin-treated extracts to PI(4,5)$P_2$-containing

supported bilayers supplemented with PI(3)P, PI(3,4)P$_2$ or PI(3,4,5)P$_3$. PI(3)P and not PI(3,4)P2 or PI(3,4,5)P$_3$ rescues FLS formation, confirming it is the active lipid (Fig. 3, C and D). The functional activity of PI(3)P in FLS may suggest that PI(3,4,5)P$_3$ and PI(3,4)P$_2$ in filopodia are dephosphorylated to produce PI(3)P for actin nucleation in filopodia, similar to our observations in endocytosis (Daste et al., 2017).

### SNX9 localizes to the tip and shaft of cellular filopodia

While FLS allow biochemical analysis, they lack a boundary membrane, so we asked whether SNX9 is a component of cellular filopodia. Filopodia can be readily visualized in open-face dorsal marginal zone explants from *Xenopus* embryos that undergo convergent extension, where they are implicated in generating the forces of tissue movement (Belmonte et al., 2016; Keller et al., 2000). We coexpressed mCherry-SNX9 and the plasma membrane marker GFP-CAAX, observing SNX9 at the tips of a subset of extending filopodia, which were frequently observed clustered together in distinct subregions of the explants, suggesting they may have specialized functional properties (Fig. 4 A, Fig. S3 A, and Video 1).

We next investigated the localization of endogenous SNX9 in human cells by indirect immunofluorescence of HeLa and retinal pigment epithelial (RPE-1) cells, using two independent commercial human anti-SNX9 antisera. The specificity of the antibodies for SNX9 was confirmed as the immunofluorescence signal was markedly reduced following siRNA-mediated SNX9 knockdown (Fig. 4 B and Fig. S3 B; Ford et al., 2018). There were similar numbers of filopodia in control and SNX9 knockdown cells (Fig. S3 C), which may be expected given that filopodia are still observed on Arp2/3 complex knockout (Suraneni et al., 2012; Wu et al., 2012). Strikingly, however, SNX9 was present in ~90% of filopodia and was present within the shaft as well as at the tip (Fig. 4, B–D).

Intriguingly, SNX9 is implicated in the filopodial capture of extracellular *Chlamydia trachomatis* by human cells (Ford et al., 2018). A correlative decrease in both filopodial number and bacterial internalization was observed when SNX9$^{-/-}$ knockout HAP1 cells were infected, a deficit that was partially overcome by sedimenting bacteria onto the cells (Ford et al., 2018). Consequently, we examined the relationship between infectious *C. trachomatis* elementary bodies (EBs), endogenous SNX9, and filopodia during the initial minutes of the host–pathogen interaction. Clusters or single infectious EBs were observed in association with the tips and shafts of SNX9-positive filopodia, which often appeared to capture EBs at a distance from the cell surface (Fig. 4 D, panels a–d; and Fig. S3, D–F). Rather than *C. trachomatis* inducing SNX9 filopodia, for example by using a SNX9-binding translocated bacterial effector, our findings suggest SNX9 is endogenously present in filopodia that are hijacked by *C. trachomatis* to drive entry. While the *C. trachomatis* entry mechanism shows some similarities with the determinants of FLS formation, it is not sensitive to either wortmannin or LY294002 (Ford et al., 2018). Thus, as is typical of pathogen subversion of target cells, it is likely that bacterial virulence factors replace or reprogram cellular factors to ensure deviation of bacterial-containing vacuoles from the canonical endocytic pathway.

Because of its known roles in responding to phosphoinositide lipids, endocytosis, and actin regulation, SNX9 is a candidate molecule integrating signaling, membrane traffic, and filopodia. Molecular differences have been observed between different filopodia, and SNX9 localization to a subset of filopodia in vivo supports that there may be molecular specializations appropriate to different cellular contexts. In early *Xenopus* development, many vesicles traffic within filopodia (Danilchik et al., 2013), suggesting that the shaft localization we obtained in fixed cells could correspond to vesicles trafficking within filopodia. These appear responsible for *C. trachomatis* entry, and from the in vitro work, also appear important in building the filopodium. In agreement with this, inhibiting endocytosis freezes filopodia dynamics (Bu et al., 2009; Gallop, 2019; Nozumi et al., 2017). Our work sheds light on the cellular mechanisms by which SNX9 is involved in development, cancer metastasis, and pathogen entry and holds promise for further molecular dissection of FLS in vitro and filopodia in vivo.

## Materials and methods

### Cell lines

RPE-1 cells (ATCC CRL-4000, RRID:CVCL 4388) are an immortalized line derived from human female retinal pigmental epithelial tissue. They were maintained in DMEM/F12 media (Sigma-Aldrich, D6421) supplemented with 10% fetal bovine serum, 0.25% sodium bicarbonate (both Sigma-Aldrich), 100 µg/ml penicillin/100 U/ml streptomycin, and 2 mM L-glutamine (all Gibco). HeLa (ATCC CCL-2, RRID:CVCL 0030) cells (derived from human cervical adenocarcinoma) were maintained in DMEM (high glucose + glutamax, Gibco) containing 10% FBS and 100 µg/ml penicillin/100 U/ml streptomycin. Both lines were grown at 37°C in a humidified incubator containing 5% CO$_2$, and were subcultured twice weekly.

### *Xenopus laevis*

*X. laevis* (RRID:XEP_Xla100) were used as both a source for the egg extracts used in FLS assays and as a model organism in *Xenopus* embryo explant experiments.

Male and female *X. laevis* were housed in the aquatic facility at the Gurdon Institute. *Xenopus* were housed in a Marine Biotech recirculating system, with 12 h light/dark cycles, and fed a commercial trout pellet diet. Female *X. laevis* were used for egg collection only. To induce superovulation, adult female *X. laevis* were injected with 150 IU pregnant mare's serum gonadotrophin–Intervet (MSD Animal Health) 4–7 d pre-experiment and 400 iU human chorionic gonadotrophin 12–18 h pre-experiment. Primed *Xenopus* were kept in Marc's modified Ringers' (MMR) solution (100 mM NaCl, 5 mM Na-Hepes, pH 7.8, 2 mM KCl, 2 mM CaCl$_2$, 1 mM MgCl$_2$, and 0.1 mM EDTA). Male *X. laevis* were used for testes extraction. Male *Xenopus* were euthanized by subcutaneous injection of MS222. In vitro fertilization was performed by swirling mashed testis through collected eggs. The resultant embryos were injected with RNA at the two-cell stage, then maintained at 14°C, with explants taken at Nieuwkoop and Faber Stage 10. This research has been regulated under the Animals (Scientific Procedures) Act 1986 Amendment Regulations 2012 after ethical

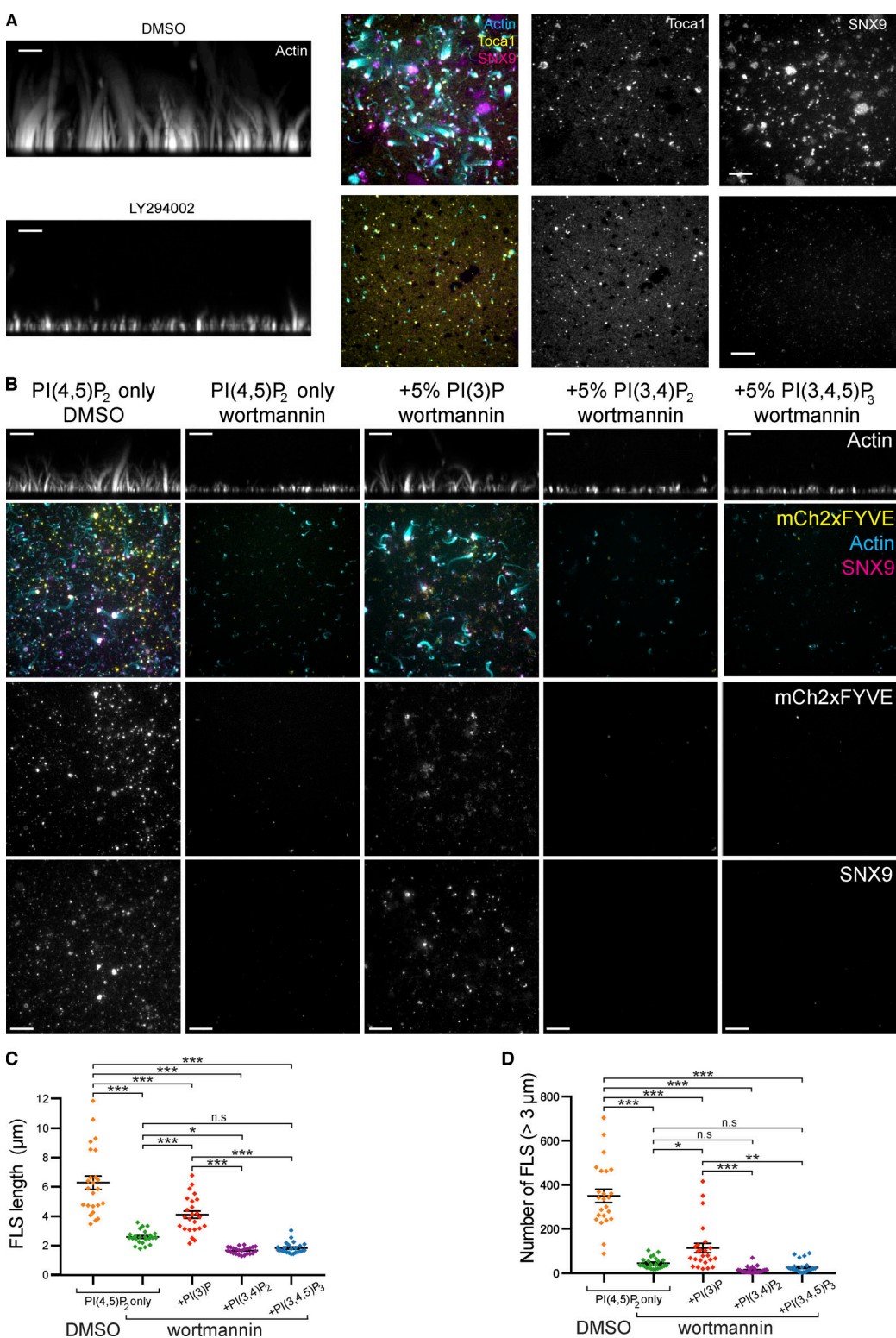

Figure 3. **Class I PI3K is involved in FLS formation with PI(3)P being a key downstream lipid. (A)** TIRF and confocal imaging of representative FLS regions containing labeled SNX9 (TIRF), TOCA-1 (TIRF), or actin (maximally projected confocal side or Z views) treated with DMSO or 100 µM LY294002. Both SNX9 and actin but not TOCA-1 are reduced by the inhibitor. **(B)** TIRF and confocal imaging of representative FLS regions containing labeled SNX9, the PI(3)P probe mCh-2xFYVE (TIRF), or actin (maximally projected confocal side or Z views) grown on membranes containing 10% PI(4,5)$P_2$ or 10% PI(4,5)$P_2$ plus 5% PI(3)P, 5% PI(3,4)$P_2$ or 5% PI(3,4,5)$P_3$ and treated with either DMSO or 2 µM wortmannin. SNX9, mCh-2xFYVE, and actin are reduced on wortmannin treatment. Inclusion of 5% PI(3)P shows mCh-FYVE labeling and SNX9 recruitment in the presence of wortmannin and 5% PI(3,4)$P_2$ or 5% PI(3,4,5)$P_3$ do not. **(C and D)** Quantification of data in B. Each datapoint represents an individual imaging region ($n$ = 24 regions for each treatment from two independent experiments). Lines indicate mean ± SEM. Statistical significance was assessed by ordinary one-way ANOVA with Holm–Sidak's multiple comparisons test (Table S2). Scale bars, 10 µm. n.s., not significant.

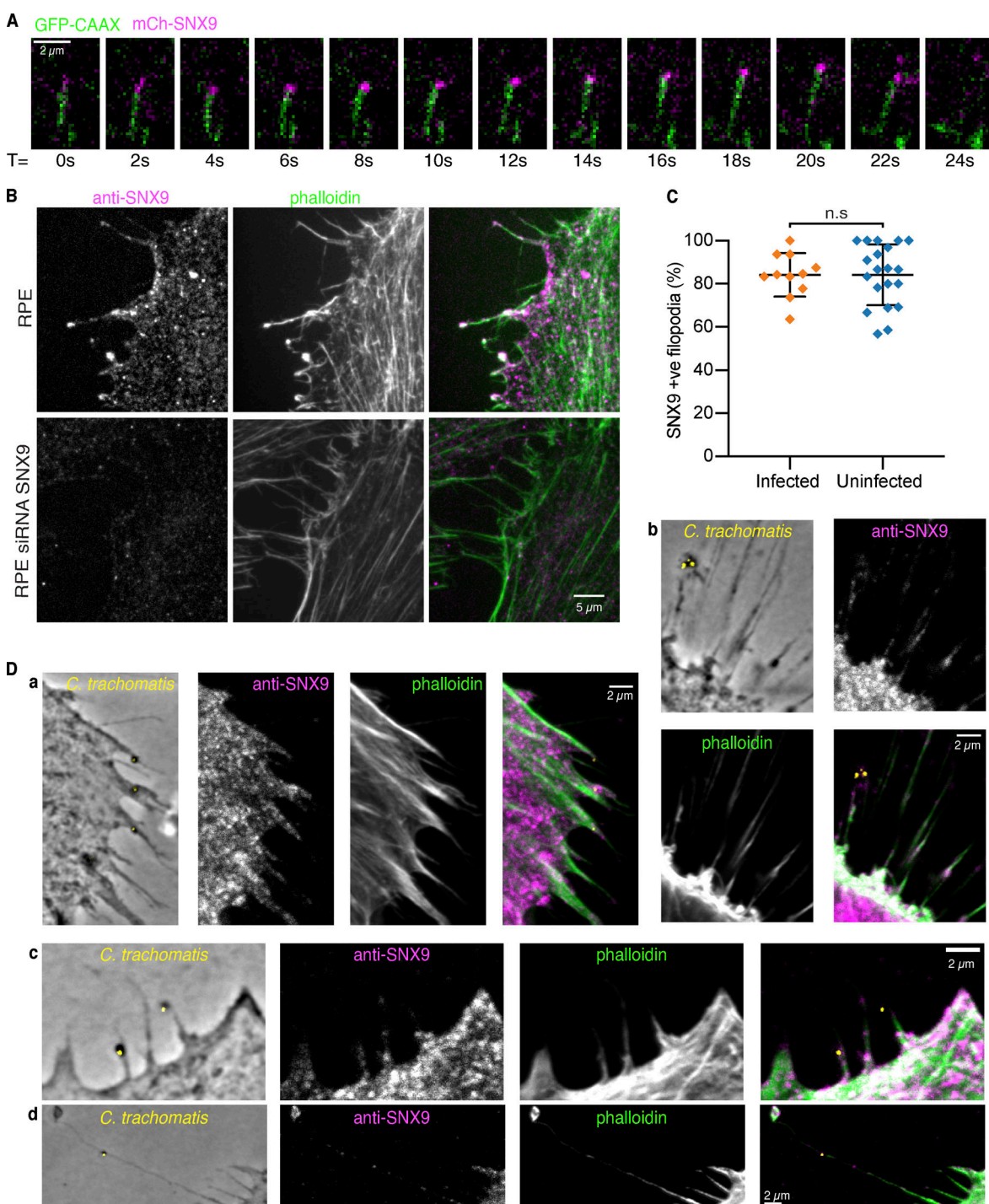

Figure 4. **SNX9 localizes to filopodia in *Xenopus* gastrula explants and human cultured cells. (A)** Background-subtracted single frames from a TIRFM time-lapse sequence taken at 2-s intervals of *Xenopus* chordamesoderm cells expressing mCh-SNX9 and GFP-CAAX membrane marker shows SNX9 tracking a growing filopodium tip. Scale bar, 2 µm. **(B)** Confocal images of RPE-1 cells treated with either scramble or SNX9 siRNA and immunolabeled with rabbit anti-human SNX9 and phalloidin; endogenous SNX9 is in RPE-1 cell filopodia and is abolished on knockdown. Scale bar, 5 µm. **(C)** Quantification of SNX9 in filopodia in uninfected RPE-1 and in cells infected with *C. trachomatis*. Each data point represents an individual imaging region; lines indicate mean ± SD. For infected cells, *n* = 248 filopodia from 62 cells across *n* = 11 images, for uninfected, *n* = 416 filopodia from 139 cells across *n* = 20 images. Unpaired *t* test, P = 0.9958 (n.s.). **(D)** a–d: Example images of phase contrast and fluorescence confocal images of RPE-1 cells infected with *C. trachomatis* and immunolabeled for SNX9 and phalloidin show *C. trachomatis* sticking to filopodia containing SNX9. Scale bars, 2 µm.

review by the University of Cambridge Animal Welfare and Ethical Review Body and covered by Home Office Project License P1B9A7D57 (license holder: J.L. Gallop) and Home Office Personal licenses held by J.L. Gallop, T. Jones-Green, and J. Mason.

### Preparation of *Xenopus* egg extracts
High-speed supernatant *Xenopus* egg extracts were prepared according to our usual methods (Walrant et al., 2015). Briefly, eggs from up to 10 *Xenopus* were dejellied in MMR containing 2% wt/vol cysteine (adjusted to pH 8.0), then gently washed in *Xenopus* extracts buffer (XB: 100 mM KCl, 100 nM CaCl$_2$, 1 mM MgCl$_2$, 50 mM sucrose, and 10 mM K-Hepes, pH 7.4), followed by extract buffer for cytostatic factor extracts (CSF-XB: 100 mM KCl, 10 mM K-Hepes, pH 7.4, 5 mM EGTA, 2 mM MgCl$_2$, and 50 mM sucrose). Following addition of protease inhibitor cocktail (Sigma-Aldrich P8340) and Energy Mix (final 1× concentration 2 mM ATP, 15 mM creatine phosphate, 2 mM MgCl$_2$), eggs were transferred to thin-wall centrifuge tubes and crushed by centrifugation at 17,800 $g$ (in a SW-40 Ti rotor [Beckman-Coulter] for 10 min at 4°C). The crude cytoplasmic extract layer was extracted using a needle and diluted 10-fold in XB, then spun at 260,000 $g$ in a Ti-70 rotor (Beckman-Coulter) for 1 h at 4°C to prepare high-speed supernatant. This supernatant was filtered through a 0.22-μm syringe filter unit, then spin-concentrated in a 10-kD molecular weight cut off (MWCO) spin concentrator (Amicon) to a final concentration of 25 mg/ml. Extracts were then supplemented with 200 mM sucrose and snap frozen in liquid nitrogen for storage at –80°C.

### Preparation of supported lipid bilayers
Supported lipid bilayers were prepared according to our usual methods (Walrant et al., 2015). Briefly, 2 mM total lipid liposomes were prepared containing, by mol fraction, 10% PI(4,5)P$_2$, 30% PS, and 60% phosphatidylcholine (Avanti Polar Lipids, 840046X, 840032C, and 840053C). For experiments that also included 5% PI(3)P, 5% PI(3,4)P$_2$, or 5% PI(3,4,5)P$_3$ (Avanti Polar Lipids, 850150P, 850153P, and 850156P), the amount of phosphatidylcholine was reduced to 55%. To prepare liposomes, lipid stocks were mixed together and dried under a constant stream of nitrogen, then under vacuum for 1 h, after which the lipid film was resuspended to the final 2 mM total lipid concentration in XB, and sonicated in a bath sonicator for 15 min. Liposomes were added to wells formed by silicone gaskets (Grace Bio-Labs, supplied by Stratech Scientific, 103280) and placed on glass coverslips or glass-bottomed Proplate Microtiter Plates (Stratech Scientific, 204969) according to the experiment, and after adsorption and fusion to the glass surface, supported bilayers were washed several times with XB.

### Phage display screening and generation of antibodies
#### Isolation of anti-FLS antibodies
Supported lipid bilayers were prepared on Proplate Microtiter Plates. FLS assays were performed according to our usual methods (Walrant et al., 2015). Briefly, assay mix comprising 6 mg/ml *Xenopus* egg extracts, 2 mM DTT, 5 μM actin, energy regenerating system (Energy Mix; final concentration 2 mM ATP, 15 mM creatine phosphate, and 2 mM MgCl$_2$) in XB was

added to the supported bilayers and incubated for 15 min at room temperature (Walrant et al., 2015). For the mature FLS condition, 2 μl phalloidin (Thermo Fisher Scientific, P3457) from a 10 mg/ml methanol stock was added and incubated for a further 5 min followed by three washes with XB and fixation with 4% formaldehyde for 75 min. For the washed FLS condition, after 15 min of FLS growth, three washes with XB were performed then the FLS stabilized by addition of 2 μl phalloidin. After a 5 min incubation, assays were washed three times with XB and fixed with 4% formaldehyde in XB for 75 min. For the early FLS condition, the assay mix was removed after 3 min and washed three times with XB. 2 μl phalloidin was added, and the mix was incubated for 5 min, then fixed with 4% formaldehyde in XB for 75 min. Fixed FLS were kept at 4°C for up to 3 d. Phenotypic selections were performed using the CAT2.0 human scFv phage display library (Lloyd et al., 2009; Vaughan et al., 1996). Three rounds of selection were performed against each of three different filopodia structures: the mature, the washed, and the early time point. Wells were washed with PBS and then blocked for 1 h with PBS-Marvel milk powder (3% wt/vol). 10$^9$–10$^{12}$ phages were blocked for 1 h at room temperature in PBS-Marvel milk powder (3% wt/vol) and then transferred to wells coated with lipid bilayers and purified FLS proteins to deplete nonspecific and known binders. The blocked phages were subsequently incubated with the appropriate FLS condition for 1 h at room temperature and any unbound phage removed by a series of wash cycles using PBS-Tween (0.1% vol/vol) and PBS. Bound phage particles were eluted by addition of 5 μg/ml trypsin, infected into *Escherichia coli* TG1 bacteria and rescued for the next round of selection (Vaughan et al., 1996).

### Phage ELISA
Following three rounds of phage display selection, individual scFv were prepared as phage supernatants, and the round 2 and 3 outputs screened by phage ELISA for binding to the corresponding filopodia structures used for selections (Osbourn et al., 1996; Vaughan et al., 1996). Nunc Maxisorp (Immobiliser) plates (Thermo Fisher Scientific, 44–2404-21) coated with lipid bilayers and a mix of purified TOCA-1, fascin, Arp2/3 complex, N-WASP, Ena, and VASP as described in (Dobramysl et al., 2019 *Preprint*) were used for negative selection. Specific clones, defined as those which gave more than threefold signal over background, were sequenced, expressed in TG1 *E. coli*, and purified via the C-terminal His tag by immobilized nickel chelate chromatography (Bannister et al., 2006; Lloyd et al., 2009).

### Reformatting of scFv to IgG1 TM
Antibodies were converted from scFv to whole immunoglobulin G1 triple mutant (IgG1-TM, IgG1 Fc sequence incorporating mutations L234F, L235E, and P331S) antibody format essentially as described by Persic et al. (1997) with the following modifications. An OriP fragment was included in the expression vectors to facilitate use with CHO-transient cells and to allow episomal replication. The variable heavy domain was cloned into a vector containing the human heavy chain constant domains and regulatory elements to express whole IgG1-TM heavy chain in mammalian cells. Similarly, the variable light domain was

cloned into a vector for the expression of the human light chain (lambda) constant domains and regulatory elements to express whole IgG light chain in mammalian cells. The plasmids were cotransfected into CHO-transient mammalian cells (Daramola et al., 2014) and IgG proteins purified from cell culture medium using Protein A chromatography. The purified IgG were analyzed for aggregation and degradation purity using size exclusion chromatography–HPLC and by SDS-PAGE.

## Cloning

*X. laevis* SNX 9 (accession no. BC077183) was PCR-amplified from IMAGE clone 3402622 (Source Bioscience) and cloned into either pCS2 N-terminal GFP or pCS2 N-terminal mCherry (primers 5′-GCATGGCCGGCCACCATGAACAGCTTTGCGG-3′ and 5′-GGCGCG CCTCACATCACTGGG-3′). GFP-Utrophin-CH was PCR amplified from a pCS2 construct (gift from S. Woolner (University of Manchester, Manchester, UK; Burkel et al., 2007) and cloned into pGEX FA acceptor vector (primers 5′-GCATGGCCGGCCACC ATGGTGAGCAAGGG-3′ and 5′-GGCGCGCCCTTGAGCTCGAGT TAGTCTATG-3′). pCS2-his-SNAP-*X. laevis* SNX9 was previously described (Daste et al., 2017). pET-his-SNAP-*Xenopus tropicalis*-Toca1 and pET-KCK-VASP were previously described (Dobramysl et al., 2019 *Preprint*). pCS2-GAP-RFP was a gift from C. Holt (University of Cambridge, Cambridge, UK). pCS2-GFP-CAAX was a gift from M. Kirschner (Harvard Medical School, Boston, MA).

## Protein purification

All chemicals and reagents were from Sigma-Aldrich unless otherwise stated. 6xHis-SNAP-TOCA-1 and 6xHis-GFP-Utrophin in pET or pGEX plasmids, respectively, were transformed into BL21 pLysS *E. coli*, with protein expression induced overnight at 19°C. The 6xHis-SNAP-SNX9 pCS2 construct was transfected into 293F cells by 293fectin reagent (Thermo Fisher Scientific) according to the manufacturer's instructions, and cells were harvested 48 h later. All purification steps were performed at 4°C. Bacteria or 293F cells were harvested by centrifugation and resuspended in a buffer containing 150 mM NaCl, 20 mM Na-Hepes, pH 7.4, 2 mM 2-mercaptoethanol, and EDTA-free cOmplete protease inhibitor tablets, after which they were lysed by probe sonication. Lysates were spun at 40,000 rpm for 45 min in a 70Ti rotor (Beckman-Coulter), with supernatants then applied to nickel nicotinamideagarose beads (Qiagen, L30210) for affinity purification. Proteins were eluted by applying stepwise increasing concentrations of imidazole (50–300 mM) in a buffer containing 150 mM NaCl, 20 mM Na-Hepes, pH 7.4, and 2 mM 2-mercaptoethanol. 6xHis-KCK-VASP in the pET15b plasmid was transformed into Rosetta DE3 pLysS cells and expression induced as for the pET constructs above. The protein was purified as above, with the following differences: all buffers contained 300 mM NaCl rather than 150 mM, and the beads used for affinity purification were cobalt agarose beads (Talon Superflow, GE Healthcare, GE28-9574-99). For all proteins, fractions containing eluted protein were further purified using S200 gel filtration on an AKTA FPLC (GE healthcare) in a buffer containing 150 mM NaCl (300 mM for KCK-VASP), 20 mM Na-Hepes, pH 7.4, 2 mM EDTA, and 5 mM DTT. Purification was verified by SDS-PAGE electrophoresis and Coomassie staining, with positive

fractions pooled and concentrated in a 10 kD MWCO spin concentrator (Millipore). Concentration was verified by A280 measurements on a Nanodrop, and proteins had 10% glycerol added before they were snap-frozen in liquid nitrogen and stored at –80°C.

## Chemical labeling of proteins with fluorescent dyes

SNAP-tagged TOCA-1 and SNX9 were labeled using SNAP-Surface Alexa Fluor 488 or Alexa Fluor 647 (New England Biolabs, S9129S, S9136S). 5–10 µM final concentration of protein was mixed with 10 µM SNAP-dye in a buffer containing 150 mM NaCl, 20 mM Na-Hepes, pH 7.4, 1 mM DTT, and 1% (vol/vol) TWEEN 20 and incubated under gentle rotation at 4°C overnight. The labeled protein was dialysed into a buffer containing 150 mM NaCl, 20 mM Na-Hepes, pH 7.4, and 10% glycerol in a 0.1-ml 20-kD MWCO Side-A-Lyzer MINI Dialysis device (Thermo Fisher Scientific) to remove excess dye; dialysis occurred in two rounds over a 24-h period. KCK-VASP was labeled at 4°C overnight by adding a 10–20 fold molar excess of the Alexa Fluor 568 maleimide dye (Thermo Fisher Scientific, A-20341) to a 50 µM final concentration of the protein in the presence of a 10-fold molar excess of TCEP under gentle rotation. Excess dye was removed by buffer exchange into 300 mM NaCl, 20 mM Na-Hepes, pH 7.4, and 10% glycerol using an Ultra-15 centrifugal filter unit spin concentrator (Amicon) with an Ultracel-10 membrane (Millipore).

## Immunodepletion

SNX9 was immunodepleted from *Xenopus* egg extracts using a rabbit anti-*Xenopus* SNX9 antibody, as previously described (Gallop et al., 2013). Three rounds of depletion were performed using Protein A beads (Thermo Fisher Scientific, 10002D). Briefly, 300 µl antibody serum (SNX9 and prebleed serum as mock sample) was bound to 75 µl Dynabead protein A beads for 1 h at room temperature with continuous rotation. Beads were washed with 1 × PBS + 0.1% Tween (PBST), PBST plus 500 mM NaCl, and XB buffer and then divided into three 25-µl aliquots. 100 µl *Xenopus* egg extracts were incubated subsequently with each aliquot of beads for 30 min at 4°C with rotation for each incubation. The resulting depleted and mock-depleted egg extracts were analyzed by Western blotting with IgFls3, IgFls4, and rabbit anti-*Xenopus* SNX9 antibody used as primary antibodies, and mouse anti-human IgG1 Fc secondary antibody, HRP secondary antibody for the IgGs, and IRDye 800CW goat anti-rabbit IgG secondary antibody for the SNX9 antibody.

## FLS assays

FLS assays were performed according to our usual methods on glass coverslips in wells defined by silicone gaskets (Walrant et al., 2015). The basic reaction mix was composed of a 1:6 dilution of 25 mg/ml *Xenopus* egg extracts, 2 mM DTT, Energy Mix, 1 µM unlabeled rabbit skeletal muscle actin (Cytoskeleton, AKL99), and 210 nM fluorescently labeled actin (Alexa Fluor 488, 568, or 647, Thermo Fisher Scientific, A12373, A12374, A34051), made up in XB, and supplemented with other labeled proteins, antibodies, or inhibitors as necessary for the specific experiment. 50 µl of reaction mix was gently added to the supported lipid bilayer and incubated at room temperature for

25 min before imaging. For the screening of the scFvs, 5 µl of the individual scFv was added to the basic reaction mix. For experiments involving the rescue of FLS prepared using mock/SNX9-immunodepleted or scFV-treated extracts, 20 nM of SNX9 was added, and rather than using fluorescently labeled actin, 33 µM of GFP-utrophin was used to label FLS (Burkel et al., 2007). For experiments in which the localization of fluorescently labeled recombinant purified proteins was analyzed, they were included in the reaction mix at the following concentrations: 20 nM VASP, 30 nM SNX9, 10 nM TOCA-1, and 1 µg/ml mCh-2xFYVE. For experiments involving the addition of inhibitors to FLS, inhibitors (or appropriate dilutions of the vehicle, DMSO) were prepared as 10× stocks in XB, with 5 µl then added to complete the 50 µl reaction mix. The mixes were then incubated for 10 min before being added to the supported lipid bilayer to start the experiment. Inhibitors were used at the following final concentrations: 2 µM wortmannin (Sigma-Aldrich, W-1628), and 100 µM LY294002 (Selleck Chemicals, S1105).

### Immunolabeling of FLS

FLS were prepared using an assay mix comprised of 6 mg/ml *Xenopus* egg extracts, 2 mM DTT, Energy Mix, and 210 nM fluorescently Alexa Fluor 568–labeled actin made up in XB, which was added to the supported bilayers and incubated for 15 min at room temperature, after which 0.5 µl phalloidin from a 10 mg/ml methanol stock was added. Following a further 5 min incubation, FLS were washed twice in XB, then fixed in 4% formaldehyde made up in cytoskeletal buffer 10 mM MES, pH 6.1, 138 mM KCl, 3 mM MgCl$_2$ and 2 mM EGTA) for 1 h. Following 3× PBS washes, FLS were blocked in 2% BSA-PBS for 1 h. Primary antibody mixes were made up in PBS; FLS were stained with the rabbit anti-*Xenopus* SNX9 antibody (1:100 dilution) for 1 h. Following 3× PBS washes, the secondary antibody mix (goat anti-rabbit Alexa Fluor 488 [Thermo Fisher Scientific, A-11008], made up in PBS) was added for 1 h. Fixed and labeled FLS were washed three further times in PBS, then immediately imaged.

### Western blotting using FLS ScFvs and IgG1-TMs

Samples of either *Xenopus* egg extracts or purified recombinant SNX9 were run on 4–20% mini-PROTEAN precast protein gels (Biorad) and transferred to nitrocellulose membrane using an iblot 2 dry blotting system. Western blots using ScFvs were blocked in 10% milk/PBST, incubated with 4.5 µg/ml ScFv in 1% milk/PBST for 1 h at room temperature, washed with PBST, and then incubated overnight with a 1/1,000 dilution in 1% milk/PBST of either rabbit anti-his antibody (Abcam, ab9108) or mouse anti-myc antibody (Roche, 11667149001). After washing the membranes, tertiary incubations were performed with a 1/10,000 dilution in 1% milk/PBST of either anti-rabbit or anti-mouse 800 CW antibody (LI-COR Biosciences, 926–32210 and 926–32211), washed, and imaged using an Odyssey Sa Reader (LI-COR Biosciences). Western blots using Fls IgGs were blocked in 5% milk/1 × TBS + 0.1% Tween. Antibody incubations were performed in antibody diluted in 5% milk/1 × TBS + 0.1% Tween. Fls IgGs were diluted to 2.5 µg/ml and the secondary mouse anti-Human IgG1-HRP antibody (Thermo Fisher Scientific, A-10648)

used at 1/500 dilution. HRP was detected using the ECL Prime Western blotting system (GE Healthcare, GERPN2232).

### Immunoprecipitation

For immunoprecipitation of IgG-FLS antigens, we employed a gravity flow column format as follows, with all steps conducted at 4°C. 2 ml of Affigel 10 matrix (BioRad, 1536099) slurry (1 ml column bed volume) was added to a poly prep chromatography column (BioRad) and washed with three bed volumes of ice-cold water under gravity flow. IgG antibody was incubated with the immunoaffinity resin at a concentration of 1 mg/ml in 20 mM Hepes, pH 7.4; 1 × PBS and columns were rotated overnight at 4°C. Affigel activated affinity resin binds to free primary amines and therefore it was necessary to block unreacted residues in order to avoid nonspecific protein binding, by the addition of 100 µl of 1 M ethanolamine (pH 8.0) spiked into the resin/antibody mix. The column was left to rotate for 1 h at 4°C. Unbound supernatant was eluted from the resin under gravity flow, and the column was washed with eight column bed volumes of 20 mM Hepes, pH 7.4. The efficiency of antibody binding to the matrix was evaluated by SDS-PAGE before proceeding to the next step. 1 ml of *Xenopus* egg extracts (∼12 mg/ml concentration) were mixed with 20 µl 10% Tween, 200 µl 10× XB, and 780 µl water and loaded onto the IgG-bound column. The columns were rotated overnight at 4°C in the presence of the antigen-containing extracts. Unbound supernatant was removed from the column under gravity flow, and the column was washed with 10 ml 0.5 M NaCl, followed by three 10-ml washes with binding buffer (20 mM Hepes, pH 7.4, 1 × PBS). All flow-through and wash samples were retained for further analysis. Elution of the antibody-bound antigen was enabled by 10 sequential additions of 250 µl elution buffer (200 mM glycine, pH 2.5, 150 mM NaCl). Each of the 10 eluates were collected into separate tubes and immediately neutralized by the addition of 50 µl 1 M Tris, pH 8.0. In total, three samples were processed in parallel: (1) immunoprecipitation with the FLS-IgG of interest, (2) immunoprecipitation with a control IgG antibody was used to assess nonspecific binding to IgG, and (3) a column with resin was prepared (as above) and incubated with extracts without the addition of IgG, to account for nonspecific binding of proteins to the resin matrix.

### Proteolytic digestion and liquid chromatography tandem mass spectrometry (LC-MS/MS)

Immunoprecipitated eluates from (1) IgFls3 IgG, (2) control IgG, and (3) *Xenopus* egg extracts + resin were concentrated by vacuum centrifugation (Labconco), run on 4–15% Mini-Protean TGX gels (Bio-Rad), and stained with Instant Blue Coomassie stain (Sigma-Aldrich). The fourth elution of the 10 sequential eluates was processed for LC-MS/MS, as this was the sample containing the antigen of interest identified by immunoblotting. Bands were excised from the Coomassie gel corresponding to the 50–80-kD region and processed for mass spectrometry as follows. Gel pieces were cut into 1–2-mm cubes, destained by several washes in 50% (vol/vol) acetonitrile, 100 mM ammonium bicarbonate solution, and then dehydrated with 100% (vol/vol) acetonitrile. Disulfide reduction was achieved by incubation

with 10 mM DTT at 37°C for 1 h, followed by alkylation of cysteine residues with 5 mM iodoacetamide at room temperature protected from light for 1 h, where both solutions were made in 100 mM ammonium bicarbonate. Gel pieces were washed several times in 50% (vol/vol) acetonitrile, 100 mM ammonium bicarbonate solution at 37°C, before being dehydrated in 100% (vol/vol) acetonitrile. Proteolytic digestion was achieved by the addition of 50 µl sequencing-grade modified trypsin (Promega) at a concentration of 10 ng/µl in 50 mM ammonium bicarbonate. A further aliquot of trypsin solution was added after 1 h to ensure gel pieces remained hydrated during overnight incubation at 37°C. The supernatant containing the eluted peptides was retained and analyzed by LC-MS/MS analysis with a Waters nanoAcquity UPLC (Thermo Fisher Scientific) system coupled to an Orbitrap Velos mass spectrometer (Thermo Fisher Scientific). Peptides were first loaded onto a precolumn (Waters UPLC Trap Symmetry C18, 180-µm internal diameter. × 20 mm, 5 µm particle size) and eluted onto a C18 reverse-phase column at a flow rate of 300 nl/min (nanoAcquity UPLC BEH C18; 75-µm internal diameter × 250 mm, 1.7 µm particle size, Waters) using a linear gradient of 3–40% buffer B over 40 min (60 min total run time including high organic wash and reequilibrium steps). Buffer A was 0.1% formic acid in HPLC-grade water (vol/vol); buffer B was 0.1% formic acid in acetonitrile (vol/vol). The mass spectrometer was operated using data-dependent tandem mass spectrometry acquisition in positive ion mode. Full scans (380–1,500 m/z) were performed in the Orbitrap with nominal resolution of 30,000. The top 20 most intense monoisotopic ions from each full scan were selected for collision-induced dissociation in the linear ion trap with a 2.0 m/z precursor ion selection window and normalized collision energy of 30%. Singly charged ions were excluded from tandem mass spectrometry, and a dynamic exclusion window of ±8 ppm for 60 s was applied.

## Xenopus embryo explants
### Capped RNA synthesis
CAAX-GFP and mCherry-SNX9 capped RNA was synthesized using the mMESSAGE mMACHINE SP6 transcription kit (Invitrogen, AM1340) from NotI linearized pCS2 CAAX-GFP and pCS2 his mCherry SNX9 plasmids. Linear DNA was cleaned up before transcription using the QIAquick PCR purification kit (Qiagen). 1 µg purified linear DNA was added to a standard mMESSAGE mMACHINE transcription reaction with the addition of 2 U/µl of Murine RNase inhibitor (New England Biolabs) and incubated for 2 h at 37°C. Unincorporated nucleotides were removed using the RNeasy mini kit (Qiagen).

### RNA injection and Keller explants
9.2 nl of a solution containing both capped RNAs at 30 ng/µl was injected into each cell of dejellied two-cell Xenopus embryos maintained in MMR containing 4% Ficoll. The injected embryos were transferred to 0.1 × MMR solution after undergoing a minimum of one cell division and cultured at 14°C. Briefly, to visualize Xenopus chordamesoderm cells, explants of dorsal mesendoderm and ectoderm were prepared from the injected embryos at the early gastrula stage (Keller and Danilchik, 1988). The vitelline membrane was removed and a section of dorsal mesendoderm and ectoderm extending from the bottle cells removed using an eyebrow knife and hair loop and cleaned of loose mesoderm cells. Explants were placed under a glass bridge in 53 mM NaCl, 32 mM Na-gluconate, 4.5 mM K-gluconate, 1 mM CaCl$_2$, 1 mM MgSO$_4$, and 5 mM Na$_2$CO$_3$, pH 8.3, with Hepes for 3 h at room temperature, and then imaged.

## Immunostaining of cells and Chlamydia infection
RPE-1 and HeLa cells were infected with wild-type C. trachomatis LGV2 as described previously (Ford et al., 2018). Briefly, cells were cultured on coverslips and infected with C. trachomatis at a multiplicity of infection of 5–30. To synchronize Chlamydia entry, bacteria were spinoculated at 900 g for 10 min. After 30 min at 37°C, samples were fixed and immunostained. For staining with mouse anti-human SNX9 antibody ([2F1], Abcam, ab118996), samples were fixed in 4% (wt/vol) paraformaldehyde in PBS for 20 min at room temperature. After fixation, samples were blocked in blocking solution (10% FCS in PBS + 0.2% [wt/vol] saponin) for 1 h, and then incubated for 1 h with primary antibody (1/100 in PBS + 3% FCS + 0.2% saponin), washed three times in PBS, and finally incubated for 1 h with goat anti-mouse IgG coupled to Alexa Fluor 568 (Thermo Fisher Scientific, A-11004, 1/500 in PBS + 3% FCS + 0.2% saponin), phalloidin coupled to Alexa Fluor 488 (Thermo Fisher Scientific, A12379, 1/100), and DAPI (Thermo Fisher Scientific, 62247, 1 µg/ml). For staining with rabbit anti-SNX9 antibody ([EPR14399] Abcam, ab181856), samples were fixed according to the "Golgi fixation method" described by Hammond et al. (2009). Samples were fixed for 15 min in 2% (wt/vol) formaldehyde in PBS. After fixation, samples were permeabilized for 5 min in permeabilization solution (20 µM digitonin, in buffer A: 150 mM NaCl, 20 mM Hepes, and 2 mM EDTA). Samples were blocked in blocking solution (10% goat serum in a buffer A) for 45 min, before incubation for 1 h with primary antibody (diluted 1:100 in buffer A + 1% goat serum), and then for 45 min with goat anti-rabbit IgG coupled to Alexa Fluor 647 (Thermo Fisher Scientific, A-21244, 1/1,000 in buffer A + 1% goat serum), phalloidin coupled to Alexa Fluor 488, and DAPI. Samples were finally fixed after staining for 5 min in 2% (wt/vol) formaldehyde solution.

## siRNA treatment
siRNA was used to knock down SNX9 expression in RPE-1 cells. Cells were seeded at an appropriate density on glass coverslips or tissue culture plates. 5 µl anti-SNX9 siRNA (5′-AACAGTCGT GCTAGTTCCTCATCCA-3′, custom sequence as used in Nández et al. (2014), synthesized by Dharmacon) or nontargeting control (Dharmacon, siGENOME nontargeting control siRNA2, D-001210-02, sequence 5′-UAAGGCUAUGAAGAGAUAC-3′). siRNA was added to 250 µl OptiMEM, with 8 µl Lipofectamine-3000 (Thermo Fisher Scientific L3000008) mixed with a second 250 µl OptiMEM aliquot. After 5 min, these were combined, and after 20 min, the siRNA complexes were added to the cells; 200 µl was added to a six-well plate well, 50 µl to a 24-well plate well. A two-shot protocol was used in which cells were transfected twice at both 24 and 72 h after seeding, with analysis (immunolabeling or Western blotting) occurring 24 h after the second shot.

## Microscopy

Imaging of all FLS experiments and live *Xenopus* explant samples were performed on a custom combined spinning disk/total internal reflection (TIRF) fluorescence microscope supplied by Cairn research. The system was based on an Eclipse Ti-E inverted microscope (Nikon), fitted with an X-Light Nipkow spinning disk (Core Research for Evolutional Science and Technology), an iLas2 illuminator (Roper Scientific), a Spectra X LED illuminator (Lumencor), and a 250-µm piezo-driven Z-stage/controller (NanoScanZ, Prior). Images were collected at room temperature through a 100× 1.49 NA oil objective using a Photometrics Evolve Delta EMCCD camera in 16-bit depth using Metamorph software (Version 7.8.2.0, Molecular Devices). Alexa Fluor 488 and GFP samples were visualized using 470/40 excitation and 525/50 emission filters, Alexa Fluor 568 and mCherry samples with 560/25 excitation and 585/50 emission filters, and Alexa Fluor 647 samples with 628/40 excitation and 700/75 emission filters. For FLS assays, specific proteins of interest at the FLS tip (at the membrane) were imaged in TIRF microscopy, in conjunction with a confocal z-stack of the actin structure. Live time-lapse *Xenopus* explant samples were imaged in TIRFM, with images captured every 2 s, at room temperature in 53 mM NaCl, 32 mM sodium gluconate, 4.5 mM potassium gluconate, 1 mM $CaCl_2$, 1 mM $MgSO_4$, 5 mM $Na_2CO_3$, and Na-Hepes, to pH 8.3). Cell culture experiments were imaged using a Zeiss LSM 700 laser scanning confocal microscope driven by Zeiss ZEN Pro software on a high resolution Axiocam camera using a Plan-Apochromat 63× 1.40 NA Oil Ph3 M27 objective. All images were processed in FIJI (ImageJ; Schindelin et al., 2012). Background-subtracted images of *Xenopus* explant time-lapse images were prepared by subtracting the average intensity of the entire time-course from each individual time point.

## Quantification and statistical analysis

Data related to FLS count and FLS physical parameters were extracted using our Fiji (ImageJ) image analysis plugin FLS Ace, described in detail in Dobramysl et al. (2019 *Preprint*). Briefly, FLS structures are segmented by applying a 2D Difference of Gaussians filter with subsequent thresholding to each z-slice. FLS positions are then traced through the stack starting at the base image by using a greedy algorithm. From this, the FLS morphology such as length, base area, and straightness are extracted.

Peptide and protein identification was conducted with the Proteome Discoverer platform (version 1.4, Thermo Fisher Scientific) using the Mascot search algorithm (version 2.6, Matrix Science) and searched against a database which combined Xenbase 9.1 and a database obtained from Wühr et al., (2014) and https://doi.org/10.17863/CAM.48349 containing 75,603 sequence entries, with the *Xenopus* accession nos. annotated by comparison to mouse orthologues (Data S1). Fixed modification of carbamidomethyl and variable modifications of oxidation and deamidation were selected. Protein grouping was enabled, a minimum ion score of 20 was required, and a strict parsimony rule was applied. Trypsin was selected as the enzyme of choice, and up to two missed cleavages were accepted. Precursor mass tolerance of 25 ppm and a fragment mass tolerance of 0.8 D were used. Percolator (version 1.17) validation was applied using a high-confidence false discovery rate threshold of 0.01 based on q value. Additional filters included a minimum protein score of 50 and a requirement of two peptides per protein group.

Presence of SNX9 in RPE-1 cell filopodia was scored by drawing a linescan region of interest over a filopodium and in an adjacent area with no filopodium in FIJI. The SD in maximal intensity value of background was calculated, and the numbers of filopodia with maximal intensity values >1 SD above background were counted.

Value and number of N, statistical tests used, and what error bars represent can be found in the figure legends. Statistical significance of results was defined as *, $P < 0.05$; **, $P < 0.01$; and ***, $P < 0.001$. The two-sample Kolmogorov–Smirnov test used in Table S1 is a four-way comparison test, with the significance thresholds divided by four by the Bonferroni correction: *, $P < 0.0125$; **, $P < 0.0025$; and ***, $P < 0.00025$. Graphs were generated and statistical analysis performed in Prism 8 (Graphpad), R, or Python (v3.6) using Matplotlib (v2.2.3) and Seaborn (v0.9.0). FLS data analysis was performed with Pandas (v0.23.4) and Scipy (v1.1.0) in a Jupyter Lab environment (v0.35.0). The tSNE plot in Fig. 1 D was generated according to the t-stochastic neighbor embedding algorithm (van der Maaten and Hinton, 2008) using the tSNE module from the scikit-learn library (v0.20.3) using six features (the median FLS count with and without added actin, the median FLS path length with and without added actin, and the median FLS base area with and without added actin) as input after preprocessing them using scikit-learn's StandardScaler module. The chosen value for the perplexity parameter was 2.49, and the value for the early exaggeration parameter was 12.

## Online supplemental material

Fig. S1 shows the FLSAce analysis outputs illustrating the effects of the addition of each scFv on FLS number and structural parameters. Fig. S2 shows the immunoprecipitation and mass spectrometry identification of IgGFls3, Western blots of scFv5 and 21 on extracts, and scFvs3,4,5 and 21 on purified SNX9, and immunofluorescence of SNX9 at FLS tips. Fig. S3 shows mNeonGreen-SNX9 at the tips of live *Xenopus* chordamesoderm cells, Western blotting illustrating SNX9 siRNA effectiveness and quantification of SNX9 knockdown on filopodia number in RPE-1 cells, and immunofluorescence images and quantification of SNX9 and *C. trachomatis* presence in filopodia in HeLa cells. Table S1 shows the statistical analysis of the data presented in Fig. 2, E and F, for the effects of scFv immunoblock and SNX9 rescue. Table S2 shows the statistical analysis of the data presented in Fig. 3, C and D, for the effects of wortmannin on FLS growth. Video 1 shows SNX9 activity at the filopodial tips of live *Xenopus* chordamesoderm cells. Data S1 is the database used to identify proteins following mass spectrometry with *Xenopus* accession nos. annotated respective to mouse orthologues.

## Acknowledgments

This work was supported by European Research Council grant 281971 and Wellcome Trust Research Career Development

Fellowship WT095829AIA to J.L. Gallop and a Medical Research Council Project Grant (N000846/1) to R.D. Hayward. We used a Wellcome Trust grant 108467/Z/15/Z multi-user equipment grant–funded mass spectrometer. We acknowledge the Gurdon Institute funding from the Wellcome Trust (092096) and Cancer Research UK (C6946/A14492). U. Dobramysl was supported by a Junior Interdisciplinary Fellowship Wellcome Trust grant no. 105602/Z/14/Z and a Herchel Smith Postdoctoral Fellowship. H. Shimo was supported by a Funai Foundation Overseas scholarship.

The authors declare no competing financial interests.

Author contributions: Conceptualization, J.L. Gallop and T.J. Vaughan; Methodology and Investigation, I.K. Jarsch, J.R. Gadsby, A. Nuccitelli, J. Mason, H. Shimo, B. Marzook, C.M. Mulvey, L. Pilloux, R.D. Hayward, and J.L. Gallop; Software and Data Curation, U. Dobramysl, C.R. Bradshaw, and C.M. Mulvey; Writing - Original Draft, J.L. Gallop, R.D. Hayward, and J.R. Gadsby; Writing - Review and Editing, C.L. Dobson, J. Mason, C.M. Mulvey, T.J. Vaughan, and B. Marzook; Supervision, J.L. Gallop, R.D. Hayward, K.S. Lilley, C.L. Dobson, and T.J. Vaughan; Project Administration, J.L. Gallop and C.L. Dobson; Funding Acquisition, J.L. Gallop.

Submitted: 8 October 2019

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

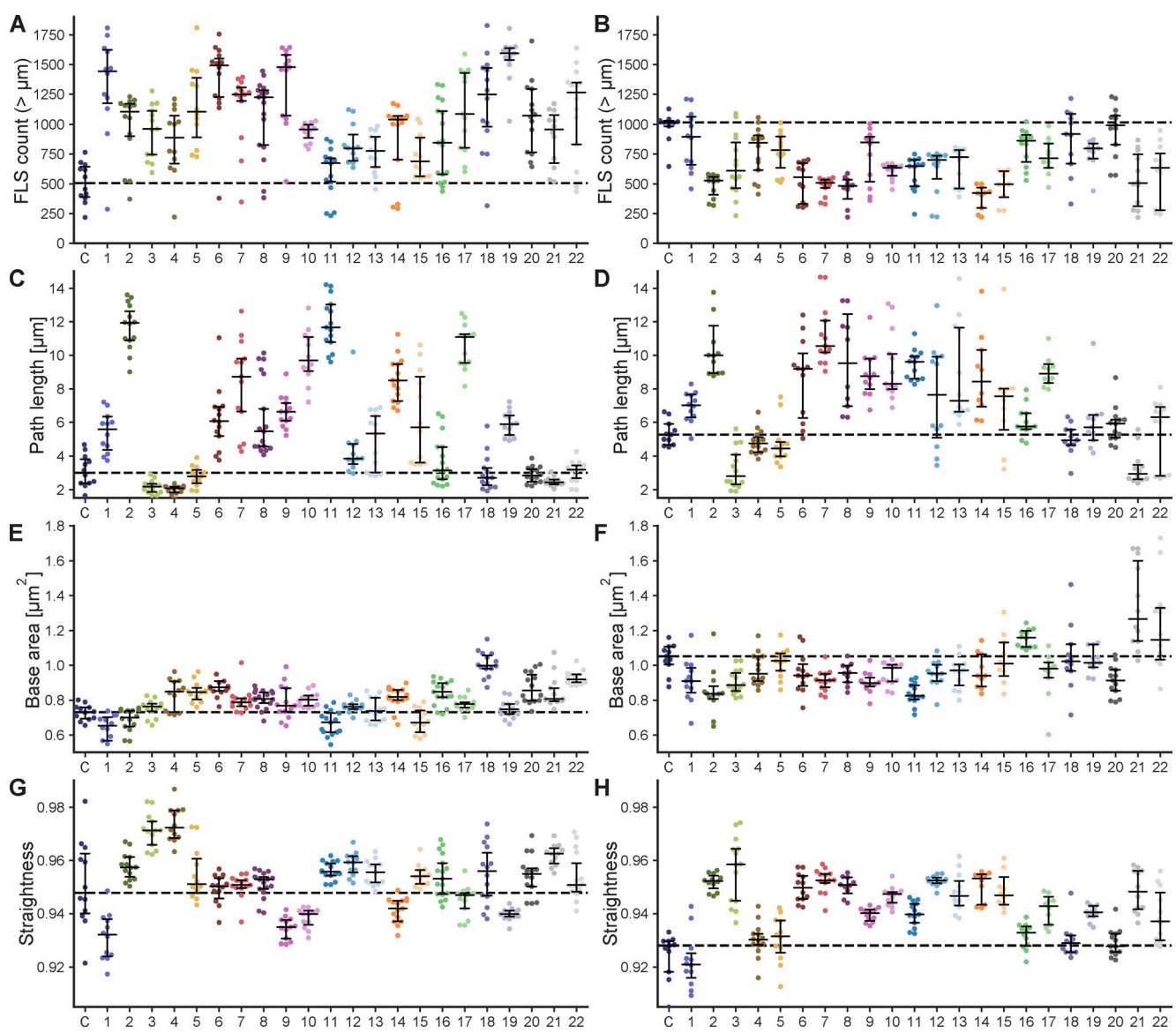

Figure S1. **Quantification of FLS phenotypes on antibody addition.** Swarmplots showing the output from FLSAce analysis for FLS assays where each different scFV is preincubated in the reaction mix, specifically illustrating the effects on number, base area (FLS thickness), length (path length) and straightness of each scFV. Colors match the colors of scFvs in Fig. 1. Data points represent mean values of FLS within an individual imaging region ($n$ = 10–17 regions per condition). Black lines indicate the median with its 95% confidence interval. Data are from three independent experiments for each scFv containing no additional actin (A, C, E, and G) and two experiments where 1 µM of supplemental unlabeled actin was added (B, D, F, and H). The addition of unlabeled actin makes FLS longer, thus increasing the length over which morphologies, such as curliness, can be seen. Additional actin also means that the phenotype of antibodies that lead to longer FLS (perhaps by inhibiting depolymerization processes) is less limited by the concentration of actin and conversely, endogenous concentrations of actin (i.e., the no additional actin condition) can make phenotypes of shorter or fewer FLS more obvious.

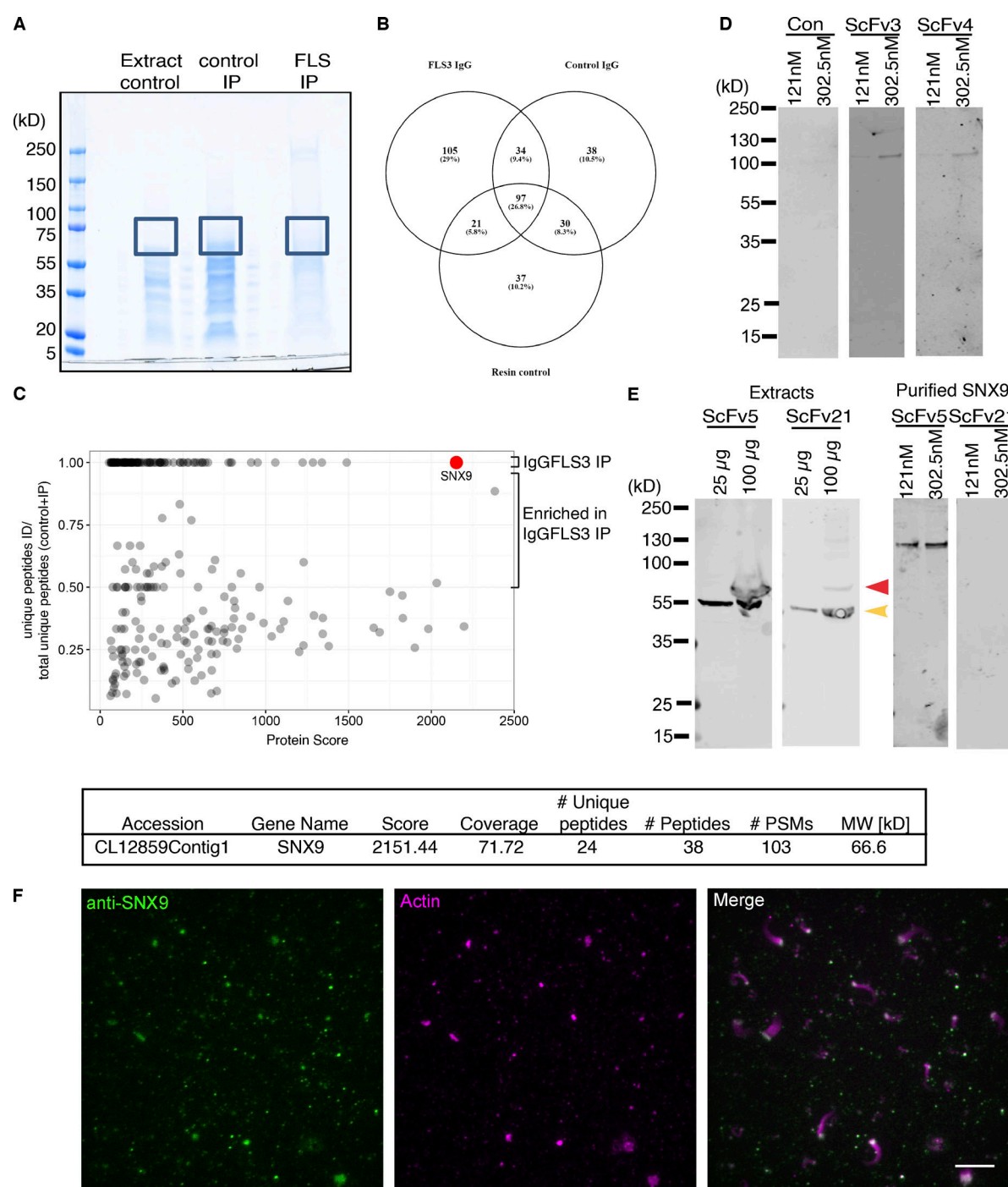

| Accession | Gene Name | Score | Coverage | # Unique peptides | # Peptides | # PSMs | MW [kD] |
|---|---|---|---|---|---|---|---|
| CL12859Contig1 | SNX9 | 2151.44 | 71.72 | 24 | 38 | 103 | 66.6 |

Figure S2. **SNX9 is the antigen to scFvs 3, 4, 5, and 21. (A)** Extracts, control, and IgGFLS3 immunoprecipitation. **(B)** Venn diagram of mass spectrometry identification from the gel sectors indicated in A. **(C)** SNX9 gives the highest protein score and number of unique peptides. **(D)** Western blot of purified SNAP-SNX9 (left lane = 121 nM, right = 302.5 nM) with scFvs 3 and 4. **(E)** Western blot of *Xenopus* egg extracts and purified SNX9 for scFv5 and 21, which detect a specific ~70 kD band (red arrow), though scFv21 does not detect purified SNX9. **(F)** TIRF and confocal imaging of FLS immunolabeled using a rabbit anti-SNX9 polyclonal antibody illustrating SNX9 (green) localized at FLS tips as marked by actin (magenta; TIRF in isolated channel, maximal projection of a 1 µm z-stack in merged image). Scale bar, 10 µm.

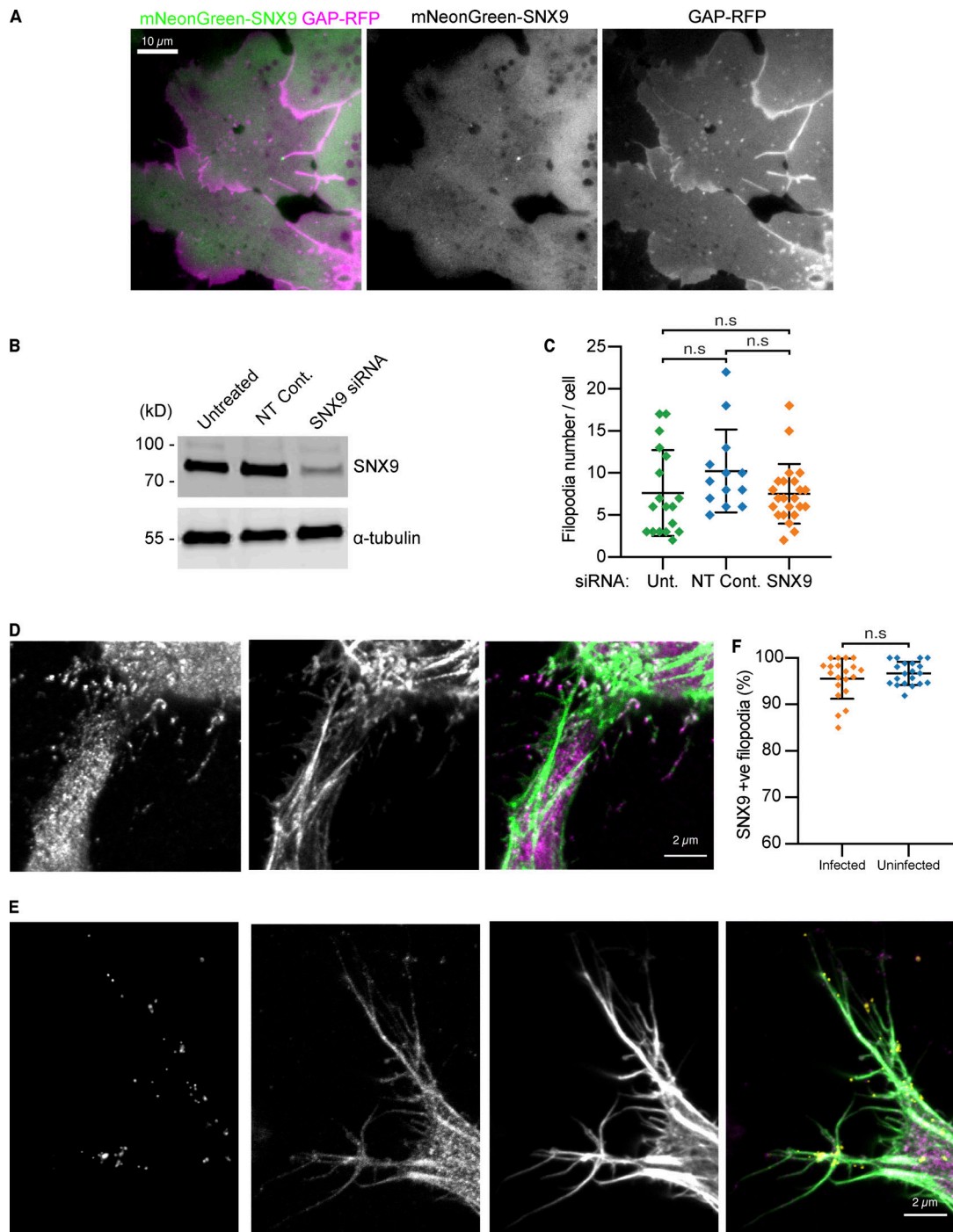

Figure S3. **SNX9 in cells. (A)** Single frame from a TIRFM time-lapse of *Xenopus* chordamesoderm cells expressing mNeonGreen-SNX9 (green) and GAP-RFP membrane marker (magenta) showing SNX9 at the tip of filopodia. Scale bar, 10 μm. **(B)** Western blots against SNX9 or the loading control DM1α (tubulin) of untreated RPE-1 cells and cells treated with either non-targeting control (Non-targ.) or anti SNX9 siRNA showing knockdown with the SNX9 siRNA sequence. **(C)** Quantification of the number of filopodia present per cell in untreated RPE-1 cells and cells treated with either nontargeting control or anti SNX9 siRNA, illustrating no change in the average number of filopodia per cell on SNX9 knockdown. Each data point represents an individual cell; *n* = 18, 13, and 23 cells for the untreated (unt.), non-targeting control (NT Cont.), and SNX9 siRNA–treated conditions, respectively. Lines indicate mean ± SD. Statistical significance was assessed by Kruskal-Wallis ANOVA with Dunn's multiple comparisons test. Overall ANOVA P = 0.1206 (n.s.), multiple comparisons: untreated vs. nontargeting control P = 0.1481 (n.s.), untreated vs. SNX9 siRNA P > 0.9999 (n.s.), nontargeting control vs. SNX9 siRNA P = 0.2800 (n.s.). **(D and E)** Example images of fluorescence confocal images of HeLa cells either (D) uninfected or (E) infected with *C. trachomatis* (yellow) and immunolabeled using mouse anti-human SNX9 (purple) and phalloidin (green), illustrating SNX9 presence in filopodia and how *C. trachomatis* stick to preexisting filopodia that contain SNX9 in either the tip or shaft of the filopodia. Scale bars, 2 μm. **(F)** Quantification of SNX9 presence in filopodia in both uninfected HeLa cells and in cells infected with *C. trachomatis*. SNX9 immunolabeling stains ∼95% of filopodia in untreated RPE-1 with no difference on *C. trachomatis* infection. Each data point represents an individual imaging region; *n* = 20 regions per condition. Lines indicate mean ± SD. Infected: 2,080 filopodia from 97 cells; uninfected: 1,648 filopodia from 126 cells. Statistical significance was assessed by unpaired *t* test, P = 0.3239 (n.s.).

Video 1. **SNX9 is active at filopodial tips in *Xenopus* dorsal marginal zone explants.** TIRFM time-lapse of *Xenopus* chordamesoderm cells expressing mNeonGreen-SNX9 (green) and GAP-RFP (magenta) showing active filopodia with SNX9 at their tips. Frames were captured every 2 s and are replayed at 15 fps. Scale bar, 10 μm.

**Tables S1 and S2 are provided online. Table S1 shows statistics for effect of immunoblock and addition of SNX9 on FLS. Quantification of the data shown in Fig. 2, E and F. Analysis uses a two-sample Kolmogorov–Smirnov test with Bonferroni correction for multiple comparisons. Imaging regions per condition: control immunoblock *n* = 10, control rescue *n* = 11, scFv3 immunoblock *n* = 10, scFv3 rescue *n* = 9, scFv4 immunoblock *n* = 9, scFv4 rescue *n* = 12, scFv5 immunoblock *n* = 12, scFv5 rescue *n* = 11, scFv21 immunoblock *n* = 12, scFv21 rescue *n* = 10. Table S2 shows statistics for effect of Wortmannin on FLS grown on different lipid compositions. Quantification of data in Fig. 3, C and D. Statistical significance for both FLS path length and FLS count (>3 μm) was assessed by ordinary one-way ANOVA with Holm-Sidak's multiple comparisons test. Overall ANOVA for both path length (Fig. 3 C) and FLS count (Fig. 3 D), ****, P < 0.0001. Significance for individual comparisons are shown in the table. *n* = 24 imaging regions for each treatment.**

**A supplemental dataset is also provided online that shows *X. laevis* proteome with *Mus musculus* orthologues.**

