## [Peer Review File · The Journal of Cell Biology]

A direct role for SNX9 in the biogenesis of filopodia

Iris Jarsch, Jonathan Gadsby, Annalisa Nuccitelli, Julia Mason, Hanae Shimo, Ludovic Pilloux, Bishara Marzook, Claire Mulvey, Ulrich Dobramysl, Charles Bradshaw, Kathryn Lilley, Richard Hayward, Tristan Vaughan, Claire Dobson, and Jennifer Gallop

Corresponding Author(s): Jennifer Gallop, University of Cambridge

Review Timeline:	Submission Date:	2019-10-08
	Editorial Decision:	2019-11-05
	Revision Received:	2020-01-24

Monitoring Editor: Pier Paolo Di Fiore

Scientific Editor: Melina Casadio

Transaction Report:

DOI: <https://doi.org/10.1083/jcb.201909178>

November 5, 2019

Re: JCB manuscript #201909178

Dr. Jennifer L Gallop
University of Cambridge
The Wellcome Trust/Cancer Research UK Gurdon Institute
Tennis Court Road
Cambridge CB2 1QN
United Kingdom

Dear Dr. Gallop,

Thank you for submitting your manuscript entitled "A direct role for SNX9 in the biogenesis of filopodia". The manuscript was assessed by expert reviewers, whose comments are appended to this letter. We invite you to submit a revision if you can address the reviewers' key concerns, as outlined here.

We sent your manuscript to experts in the fields covered by the work - trafficking, actin dynamics, and filopodia/cell migration. Although some of the referees bring up concerns about the novelty of linking SNX9 to filopodia - concerns that we also discussed editorially at submission -- you will see that they found the screening approach elegant and results important to demonstrate the success of the approach; they also all found the work of high quality and results interesting for the field. In their view (and in ours), little experimentation is needed to strengthen the current conclusions. JCB Reports do not require the same level of mechanistic understanding and depth as full Articles, and given the interest in the work, and the elegance and novelty of the screen, we would not require a major mechanistic extension of these studies (as suggested by Rev#3) for publication. We suggest that you focus efforts in revision to:

- Answer the points of Rev#1 and Rev#2 in the text - to clarify/streamline the text and improve the discussion
- Discuss the relevance of the findings to the various types of filopodia as per Rev#3, which seems important in the context of current research in the field
- If you already have data along these lines or wish to carry out these experiments, we feel that testing whether SNX9 modulates the length/dynamics of filopodia would add value to the study. However, we would not require data addressing this question for acceptance and will leave it to you to decide whether these experiments are within the scope of the work and possible. In other words, we would be ready to move forward with acceptance if these experiments were not done and the other points above were satisfactorily addressed.

Please let us know if you have any questions or concerns - we would be happy to discuss the revisions further as needed.

While you are revising your manuscript, please also attend to the following editorial points to help expedite the publication of your manuscript. Please direct any editorial questions to the journal office. To avoid unnecessary delays in the acceptance and publication of your paper, please read the following information carefully.

1) Text and figure limits: Character count for Reports is < 20,000, not including spaces. Count includes title page, abstract, introduction, results, discussion, acknowledgments, and figure legends. Count does not include materials and methods, references, tables, or supplemental legends. Reports can have up to 5 main and 3 supplemental figures. In addition, up to 10 videos are allowed as supplemental files

2) Reports must have a combined Results and Discussion section. Please be sure to remove the "Discussion" header at resubmission and edit the text accordingly.

3) Titles, eTOC: Please consider the following revision suggestions aimed at increasing the accessibility of the work for a broad audience and non-experts.

Running title: Phage display phenotype screen shows role for SNX9 in filopodia
(we can accommodate the extension and edit for you in the system as needed)

eTOC summary: A 40-word summary that describes the context and significance of the findings for a general readership should be included on the title page. The statement should be written in the present tense and refer to the work in the third person.

- Please include a summary statement on the title page of the resubmission. It should start with "First author name(s) et al..." to match our preferred style.

4) Figure formatting: Scale bars must be present on all microscopy images, including inset magnifications. Please add scale bars to 3B (top panels)

Molecular weight or nucleic acid size markers must be included on all gel electrophoresis. Please add molecular weight with unit labels on the following panels: S2B please add unit labels

5) Statistical analysis: Error bars on graphic representations of numerical data must be clearly described in the figure legend. The number of independent data points (n) represented in a graph must be indicated in the legend. Statistical methods should be explained in full in the materials and methods. For figures presenting pooled data the statistical measure should be defined in the figure legends.

Please indicate n/sample size/how many experiments the data are representative of: 1B, 2EF

6) Materials and methods: Should be comprehensive and not simply reference a previous publication for details on how an experiment was performed. Please provide full descriptions in the text for readers who may not have access to referenced manuscripts.

- Please include database/vendor IDs for all plasmids, strains and cell lines (e.g., Addgene, ATCC, etc.) - even if described in other published work or gifted to you by other researchers. If they are not available, please detail the basic genetic features, even if previously described in other work/gifts.

- Please include the non-targeting control siRNA sequence, if made available to you from the manufacturer.

- Microscope image acquisition: The following information must be provided about the acquisition and processing of images:

a. Make and model of microscope

b. Type, magnification, and numerical aperture of the objective lenses

c. Temperature

d. imaging medium

e. Fluorochromes

f. Camera make and model

g. Acquisition software

h. Any software used for image processing subsequent to data acquisition. Please include details and types of operations involved (e.g., type of deconvolution, 3D reconstitutions, surface or volume rendering, gamma adjustments, etc.).

7) References: There is no limit to the number of references cited in a manuscript. References should be cited parenthetically in the text by author and year of publication. A

- Please abbreviate the names of journals according to PubMed.

- Please note our formatting guidelines for preprints and please be sure to reformat the following ref -- ****both for the in-text citation and reference list citation****:

<http://jcb.rupress.org/reference-guidelines>

"Dobramysl, U., I.K. Jarsch, H. Shimo, Y. Inoue, B. Richier, J.R. Gadsby, J. Mason, A. Walrant, R. Butler, E. Hannezo, B.D. Simons, and J.L. Gallop. 2019. Constrained actin dynamics emerges from variable compositions of actin regulatory protein complexes. *BioRxiv*. 525725."

8) A summary paragraph of all supplemental material should appear at the end of the Materials and methods section.

- Please include ~1 brief descriptive sentence per item.

A. MANUSCRIPT ORGANIZATION AND FORMATTING:

Full guidelines are available on our Instructions for Authors page, <http://jcb.rupress.org/submission-guidelines#revised>. ****Submission of a paper that does not conform to JCB guidelines will delay the acceptance of your manuscript.****

B. FINAL FILES:

-- High-resolution figure and video files: See our detailed guidelines for preparing your production-ready images, <http://jcb.rupress.org/fig-vid-guidelines>.

****It is JCB policy that if requested, original data images must be made available to the editors. Failure to provide original images upon request will result in unavoidable delays in publication. Please ensure that you have access to all original data images prior to final submission.****

****The license to publish form must be signed before your manuscript can be sent to production. A link to the electronic license to publish form will be sent to the corresponding author only. Please take a moment to check your funder requirements before choosing the appropriate license.****

Our typical timeframe for revisions is three months; if submitted within this timeframe, novelty will

not be reassessed at the final decision. Please note that papers are generally considered through only one revision cycle, so any revised manuscript will likely be either accepted or rejected.

Thank you for this interesting contribution to the Journal of Cell Biology. You can contact us at the journal office with any questions, cellbio@rockefeller.edu or call (212) 327-8588.

Sincerely,

Pier Paolo Di Fiore, MD, PhD
Editor, Journal of Cell Biology

Melina Casadio, PhD
Senior Scientific Editor, Journal of Cell Biology

Reviewer #1 (Comments to the Authors (Required)):

The authors use an elegant xenopus extract in vitro system for assembly of filopodia-like structures (FLS) as 'bait' to select single chain variable region fragments (scFVs) from a phage display library that bind to FLSs at different stages of assembly and denaturation. They then express and purify the scFVs and perform a 'phenotypic screen' showing that when preincubated with xenopus extracts the various scFVs perturb FLS formation in multiple ways. Focusing on a set of scFVs that inhibit early stages of FLS growth they identify SNX9 as a common antigen. Depletion experiments establish that SNX9 is required for FLS assembly and studies in xenopus embryos and mammalian tissue cultures cells confirm that SNX9 is localized to filopodia. Finally, they confirm recent findings of others that SNX9 is localized to filopodia involved in capture of pathogenic *C. trachomatis*. The strength of the paper is the novelty of the scFv screen, which was successful in identify SNX9 as an important player in early FLS assembly. However, other aspects are less novel. For example, a role for SNX9 in actin-based membrane protrusions (invadopodia, membrane ruffles, etc. has been previously demonstrated) as has a role for SNX9 in filopodia-dependent capture of *C. trachomatis*. There are other specific issues that should be addressed.

1. It is somewhat unclear what criteria were used to distinguish phenotypes caused by the various scFVs both based on the appearance of the FLS (Fig. 1C and the quantitative analyses, Fig. S1). As two examples (although there are others) scFv16 is described as having 'no phenotype', while scFv20 is described as having 'higher numbers': this is not evident either in the image or in the quantification Fig. S1A. Both svFv13 and 15 are described as having 'no phenotype' yet the quantitative data in Fig S1A shows that they behave very different from buffer controls when additional actin is present. As the screen is a novelty the authors should at least take more time to discuss this data. Perhaps including the quantitation in the main text.

2. scFv21 may not bind SNX9. Contrary to the statement (page 7) that scFV21 dependent inhibition of FLS was "similarly relieved" by SNX9, the data shows otherwise. I would be careful about this conclusion.

3. Given that PI34P2 and PI345P3 are the lipids implicated in filopodia formation, it is somewhat surprising that only PI3P, but not these other species can rescue the wortmannin treatment. The authors should discuss this.

4. While the authors have clearly shown that SNX9 is localized to filopodia and also to filopodia involved in the capture of *C. trachomatis*, I don't think their data (nor the findings of Ford et al) allows them to conclude (as they do on pg 10) that "SNX9 has a specific role in a filopodia mechanism... and in specialized filopodia...."

Minor:

Page 6 last paragraph, the sentence needs rearrangement as scFVs don't have avidity, rather they were converted to IgGs so that they can. T

Page 7 first paragraph, the sentence incorrectly states that 'the monoclonal antibodies scFv3 etc. were used for immunoblotting'.

Reviewer #2 (Comments to the Authors (Required)):

Review of "A direct role for SNX9 in the biogenesis of filopodia" by Jarsch et al.

Much of cell biology research is focused on the identification of important participants in cellular processes. Currently, favored approaches include depletion or knockout of candidate proteins followed by phenotypic analysis. These approaches can have at least one major shortcoming: pleiotropic effects if the target in question participates in many processes. In the current study, Jarsch et al. describe a new approach (or at least previously existing approaches combined in a new way) for identification of players important for filopodia generation: phage display screening to isolate antibodies that alter formation of filopodia-like structures (FLS) in *Xenopus* egg extracts formed on supported lipid bilayers. Using this approach they convincingly demonstrate that sorting nexin 9 (SNX9) is important for FLS formation and then use a variety of complementary follow-up experiments to confirm their in vitro findings in vivo.

This study is appealing for two reasons: first, the approach could be expanded to any of the many processes that can be recapitulated in *Xenopus* egg extracts-etc. spindle assembly, nuclear envelope assembly, cell cycle progression, *Listeria* rocketing and so forth. Second, it consolidates a variety of disparate observations from the literature hinting that SNX9 may be an important participant in formation of filopodia.

The only feature of this work that needs more effort is the writing. In many paragraphs, awkward or confusing sentences make it difficult to follow the logic underlying the study. In the Introduction, for example, there is no transition between the first two sentences; the authors write "What we know so far about how filopodia form remains poorly understood"-This should be "The means by which filopodia form remains poorly understood"; formins are nucleators, not nucleation pathways; the sentence "Ena/VASP proteins are important in filopodia formation in cortical neurons and terminal arborization in retinal ganglion cells, while in osteosarcoma cells although reducing levels of Ena/VASP proteins inhibits filopodia, Ena/VASP localize to focal adhesions in the cell body rather than at filopodia tips (Young et al., 2018) is so convoluted that it must be read several times to extract the meaning.

Because the approach is novel, precision in the writing is particularly important. For example, the following sentence,

"In this approach human single chain Fv fragments (scFv) are displayed on the surface of

bacteriophage and used to select antibodies engaged in specific interactions with the sample of interest"

implies that the Fv fragments are selecting the antibodies. However, it is my understanding that the Fv fragments are the antibodies.

Another example: "We excluded phage that bound under all three conditions because they were more likely to bind residual proteins from the extracts".

What are the three conditions? The authors referred to three FLS types in the preceding paragraph-are these the conditions? If so, it would be better to refer to them as types rather than conditions in the above sentence.

Some of the writing also undercuts the points made by the authors. For example, they write,

"These data reinforce the previous functional data with SNX9^{-/-} knockout cells that implicated SNX9 as a cellular mediator of filopodia formation (Ford et al., 2018)." As written, this could be taken to mean that Ford et al. showed that SNX9 is an important participant in all filapodia when in fact they reported that it is recruited to filapodia in response to Chlamydia, a point refuted by the current study.

More generally, the introduction seems like it is leading up to the conclusion that using their novel approach, the authors have identified a key player in all filapodia (as contrasted to Arp2/3 or formins). But this does not seem to be the case in that they report that it is only found in a subset of the filapodia within the *Xenopus* marginal zone explants. Unless they mean that only a subset showed tip localization (again, the writing is not clear).

Other comments:

Any statistical differences should be shown in figure 2F. Statistics should be shown for

Labels are needed for the panels in figure 4d.

Labels for fig 4 d

Reviewer #3 (Comments to the Authors (Required)):

In this manuscript Jarsch et al use a phage display phenotypic screening approach to identify antibodies that modulates filopodia formation in vitro. Using this very elegant approach the authors identified SNX9 as novel a novel filopodia component. The work is well executed, controlled and is very convincing. While this reviewer strongly believe that an unbiuous characterisation of filopodia components would be of high interest, this work only provides a list of antibody names (in addition to SNX9) that modulate filopodia in vitro. In addition, the screening approach described here, while very elegante, would be rather difficult for others to implement. Finally, while it is clear that SNX9 localise to filopodia, SNX9 does not appear to contribute to filopodia function in cells (in the conditions tested by the authors). Altogether the advances reported in this manuscript are rather limited and this article may be more suited for a more specialised journal.

Mains comments:

The authors refer to previous work to validate that SNX9 contribute to human pathogen entry. In their experiments, does SNX9 filopodia also contribute to pathogen infections ?

Regarding other filopodia functions, does SNX9 modulates filopodia length and or dynamics ? Could SNX9 modulate exosomes capture ?

From recent work from both the Higgs (Young et al 2018) and Ivaska (Jacquemet et al 2019) laboratories, it is becoming clear that different "types" of filopodia exist in cells and that these filopodia types have overlapping but distinct compositions and functions. For instance, filopodia induced by FMNL3 or MYO10 appear to be very different from each other. For instance, VASP only localise to MYO10 induced filopodia and not to FMNL3 filopodia (Young et al 2018). With this in mind I have the following suggestions:

1) In the current form of the manuscript, the authors assume that their screening approach model all types of filopodia but this may not be the case and this could be discussed.

2) Does SNX9 localise to both FMNL3 and MYO10 induced filopodia ? If not, could SNX9 regulate the functions of specific types of filopodia ?

The connection between filopodia and endocytosis is very intriguing. Can the authors observed other part of the endocytic machinery in filopodia (tip / base ?) or even detect endocytic events in these structures ?

Reviewer #1 (Comments to the Authors (Required)):

The authors use an elegant xenopus extract in vitro system for assembly of filopodia-like structures (FLS) as 'bait' to select single chain variable region fragments (scFVs) from a phage display library that bind to FLSs at different stages of assembly and denaturation. They then express and purify the scFVs and perform a 'phenotypic screen' showing that when preincubated with xenopus extracts the various scFVs perturb FLS formation in multiple ways. Focusing on a set of scFVs that inhibit early stages of FLS growth they identify SNX9 as a common antigen. Depletion experiments establish that SNX9 is required for FLS assembly and studies in xenopus embryos and mammalian tissue cultures cells confirm that SNX9 is localized to filopodia. Finally, they confirm recent findings of others that SNX9 is localized to filopodia involved in capture of pathogenic *C. trachomatis*. The strength of the paper is the novelty of the scFv screen, which was successful in identify SNX9 as an important player in early FLS assembly.

We thank the reviewer for recognizing the novelty and power of our approach.

However, other aspects are less novel. For example, a role for SNX9 in actin-based membrane protrusions (invadopodia, membrane ruffles, etc. has been previously demonstrated) as has a role for SNX9 in filopodia-dependent capture of *C. trachomatis*. There are other specific issues that should be addressed.

1. It is somewhat unclear what criteria were used to distinguish phenotypes caused by the various scFVs both based on the appearance of the FLS (Fig. 1C and the quantitative analyses, Fig. S1). As two examples (although there are others) scFv16 is described as having 'no phenotype', while scFv20 is described as having 'higher numbers': this is not evident either in the image or in the quantification Fig. S1A. Both svFv13 and 15 are described as having 'no phenotype' yet the quantitative data in Fig S1A shows that they behave very different from buffer controls when additional actin is present. As the screen is a novelty the authors should at least take more time to discuss this data. Perhaps including the quantitation in the main text.

We thank the reviewer for prompting us to improve the data analysis, we are much happier with the revision. The screen resulted in multiple quantitative measures of phenotype, with and without additional actin and we had endeavored to simplify the phenotypes and classify them by the dominant characteristic. However due to the complexity of the phenotypes this was both difficult to do and doesn't capture all the data, as the reviewer pointed out. We have now used a t-distributed stochastic neighbor embedding algorithm to convert our multi dimensional data to two dimensions (revised Fig. 1D). This results in several clusters of antibodies that have similar, complex, effects which can be readily matched to the images and bar charts in Fig. S1D. We thank the reviewer as this also provides us a map to navigate the effects of the antibodies.

2. scFv21 may not bind SNX9. Contrary to the statement (page 7) that scFV21 dependent inhibition of FLS was "similarly relieved" by SNX9, the data shows otherwise. I would be careful about this conclusion.

We agree with the reviewer's assessment and have now suggested that the antigen is likely to be a related protein.

3. Given that PI34P2 and PI345P3 are the lipids implicated in filopodia formation, it is somewhat surprising that only PI3P, but not these other species can rescue the wortmannin treatment. The authors should discuss this.

We have added new text discussing that PI(3,4)P₂ and PI(3,4,5)P₃ may be dephosphorylated to PI(3)P to activate SNX9 in filopodia (similar to our findings in endocytosis).

4. While the authors have clearly shown that SNX9 is localized to filopodia and also to filopodia involved in the capture of *C. trachomatis*, I don't think their data (nor the findings of Ford et al) allows them to conclude (as they do on pg 10) that "SNX9 has a specific role in a filopodia mechanism... and in specialized filopodia...."

In revision, we instead conclude:

Molecular differences have been observed between different filopodia and SNX9 localization to a subset of filopodia *in vivo* supports that there may be molecular specializations appropriate to different cellular contexts. In early *Xenopus* development, many vesicles traffic within filopodia (Danilchik et al., 2013) suggesting that the shaft localization we obtained in fixed cells could correspond to vesicles trafficking within filopodia. These appear responsible for *C. trachomatis* entry, and from the *in vitro* work, also appear important in building the filopodium. In agreement with this, inhibiting endocytosis freezes filopodia dynamics (Gallop, 2019; Nozumi et al., 2017). Our work sheds light on the cellular mechanisms by which SNX9 is involved in development, cancer metastasis and pathogen entry and holds promise for further molecular dissection of FLS *in vitro* and filopodia *in vivo*.

Minor:

Page 6 last paragraph, the sentence needs rearrangement as scFVs don't have avidity, rather they were converted to IgGs so that they can. T

Page 7 first paragraph, the sentence incorrectly states that 'the monoclonal antibodies scFv3 etc. were used for immunoblotting'.

Now corrected, many thanks for spotting these points.

Reviewer #2 (Comments to the Authors (Required)):

Review of "A direct role for SNX9 in the biogenesis of filopodia" by Jarsch et al.

Much of cell biology research is focused on the identification of important participants in cellular processes. Currently, favored approaches include depletion or knockout of candidate proteins followed by phenotypic analysis. These approaches can have at least one major shortcoming: pleiotropic effects if the target in question participates in many processes. In the current study, Jarsch et al. describe a new approach (or at least previously existing approaches combined in a new way) for identification of players important for filopodia generation: phage display screening to isolate antibodies that alter formation of filopodia-like structures (FLS) in *Xenopus* egg extracts formed on supported lipid bilayers. Using this approach they convincingly demonstrate that sorting nexin 9 (SNX9) is important for FLS formation and then use a variety of complementary follow-up experiments to confirm their *in vitro* findings *in vivo*.

This study is appealing for two reasons: first, the approach could be expanded to any of the many processes that can be recapitulated in *Xenopus* egg extracts- etc. spindle assembly, nuclear envelope assembly, cell cycle progression, Listeria rocketing and so forth. Second, it consolidates a variety of disparate observations from the literature hinting that SNX9 may be an important participant in formation of filopodia.

We thank the reviewer for recognizing the importance of our work both technically and conceptually.

The only feature of this work that needs more effort is the writing. In many paragraphs, awkward or confusing sentences make it difficult to follow the logic underlying the study. In the Introduction, for example, there is no transition between the first two sentences; the authors write "What we know so far about how filopodia form remains poorly understood"-This should be "The means by which filopodia form remains poorly understood"; formins are nucleators, not nucleation pathways; the sentence "Ena/VASP proteins are important in filopodia formation in cortical neurons and terminal arborization in retinal ganglion cells, while in osteosarcoma cells although reducing levels of Ena/VASP proteins inhibits filopodia, Ena/VASP localize to focal adhesions in the cell body rather than at filopodia tips (Young et al., 2018) is so convoluted that it must be read several times to extract the meaning.

We thank the reviewer for their advice. We have worked hard on the writing, together with reducing the number of words and amalgamating the results and discussion and think this version is now much clearer.

Because the approach is novel, precision in the writing is particularly important. For example, the following sentence,

"In this approach human single chain Fv fragments (scFv) are displayed on the surface of bacteriophage and used to select antibodies engaged in specific interactions with the sample of interest"

implies that the Fv fragments are selecting the antibodies. However, it is my understanding that the Fv fragments are the antibodies.

Another example: "We excluded phage that bound under all three conditions because they were more likely to bind residual proteins from the extracts".

What are the three conditions? The authors referred to three FLS types in the preceding paragraph-are these the conditions? If so, it would be better to refer to them as types rather than conditions in the above sentence.

Now corrected.

Some of the writing also undercuts the points made by the authors. For example, they write,

"These data reinforce the previous functional data with SNX9^{-/-} knockout cells that implicated SNX9 as a cellular mediator of filopodia formation (Ford et al., 2018)." As written, this could be taken to mean that Ford et al. showed that SNX9 is an important participant in all filapodia when in fact they reported that it is recruited to filapodia in response to Chlamydia, a point refuted by the current study.

More generally, the introduction seems like it is leading up to the conclusion that using their novel approach, the authors have identified a key player in all filapodia (as contrasted to Arp2/3 or formins). But this does not seem to be the case in that they report that it is only found in a subset of the filapodia within the *Xenopus* marginal zone explants. Unless they mean that only a subset showed tip localization (again, the writing is not clear).

In RPE and HeLa cells SNX9 was found in 90% of filopodia either in the shaft or tip. In the primary frog cells the role of SNX9 appeared more specialised.

Other comments:

Any statistical differences should be shown in figure 2F. Statistics should be shown for

Labels are needed for the panels in figure 4d.

Labels for fig 4 d

Thank you, we have added them.

Reviewer #3 (Comments to the Authors (Required)):

In this manuscript Jarsch et al use a phage display phenotypic screening approach to identify antibodies that modulates filopodia formation in vitro. Using this very elegant approach the authors identified SNX9 as novel a novel filopodia component. The work is well executed, controlled and is very convincing.

We thank the reviewer for pointing out the rigor and strong execution of our work.

While this reviewer strongly believe that an unbiased characterisation of filopodia components would be of high interest, this work only provides a list of antibody names (in addition to SNX9) that modulate filopodia in vitro. In addition, the screening approach described here, while very elegate, would be rather difficult for others to implement. Finally, while it is clear that SNX9 localise to filopodia, SNX9 does not appear to contribute to filopodia function in cells (in the conditions tested by the authors).

The reviewer refers to the siRNA knockdown of SNX9 not affecting filopodia numbers. A closer analysis of filopodia dynamics would be needed to make a conclusion of lack of function. Ford et al observed a deficit in Chlamydia infection, which may well be due to a subtle change in the mechanism used to form filopodia in the SNX9 KO. Such investigations are a topic of future work.

Altogether the advances reported in this manuscript are rather limited and this article may be more suited for a more specialised journal.

Mains comments:

The authors refer to previous work to validate that SNX9 contribute to human pathogen entry. In their experiments, does SNX9 filopodia also contribute to pathogen infections ?

Regarding other filopodia functions, does SNX9 modulates filopodia length and or dynamics ? Could SNX9 modulate exosomes capture ?

From recent work from both the Higgs (Young et al 2018) and Ivaska (Jacquemet et al 2019) laboratories, it is becoming clear that different "types" of filopodia exist in cells and that these filopodia types have overlapping but distinct compositions and functions. For instance, filopodia induced by FMNL3 or MYO10 appear to be very different from each other. For instance, VASP only localise to MYO10 induced filopodia and not to FMNL3 filopodia (Young et al 2018). With this in mind I have the following suggestions:

1) In the current form of the manuscript, the authors assume that their screening approach model all types of filopodia but this may not be the case and this could be discussed.

2) Does SNX9 localise to both FMNL3 and MYO10 induced filopodia ? If not, could SNX9 regulate the functions of specific types of filopodia ?

We have now discussed different types of filopodia and that there may be a

specialized vesicle traffic/filopodia link.

The connection between filopodia and endocytosis is very intriguing. Can the authors observe other parts of the endocytic machinery in filopodia (tip / base ?) or even detect endocytic events in these structures ?

Indeed, this is very intriguing, and others have reported similar observations in cellular filopodia, with endocytic proteins localizing to the tips and base. Two main studies are Bu et al (JCB 2009) which links endocytosis and filopodia via TOCA-1 and N-WASP, showing that interventions that inhibit endocytosis inhibit filopodia. Nozumi et al (Cell Reports 2017) have seen endophilin-dependent endocytic vesicles budding at filopodia and also see similar effects of dynamin inhibition. I have recently reviewed this area (Gallop, Semin Cell Dev Biol 2019).

Editorial points

- 1) Text and figure limits: Character count for Reports is < 20,000, not including spaces. Count includes title page, abstract, introduction, results, discussion, acknowledgments, and figure legends. Count does not include materials and methods, references, tables, or supplemental legends. Reports can have up to 5 main and 3 supplemental figures. In addition, up to 10 videos are allowed as supplemental files
- 2) Reports must have a combined Results and Discussion section. Please be sure to remove the "Discussion" header at resubmission and edit the text accordingly.

We have amalgamated the results and discussion sections, and made other sections of the text more clear and concise, ensuring that the total character count is <20,000.

- 3) Titles, eTOC: Please consider the following revision suggestions aimed at increasing the accessibility of the work for a broad audience and non-experts.

Running title: Phage display phenotype screen shows role for SNX9 in filopodia (we can accommodate the extension and edit for you in the system as needed)

eTOC summary: A 40-word summary that describes the context and significance of the findings for a general readership should be included on the title page. The statement should be written in the present tense and refer to the work in the third person.

- Please include a summary statement on the title page of the resubmission. It should start with "First author name(s) et al..." to match our preferred style.

We have added the running title and a summary statement to the title page. Please note that the running title will need extending as you indicated as is too long to be added to the online submission.

4) Figure formatting: Scale bars must be present on all microscopy images, including inset magnifications. Please add scale bars to 3B (top panels) Molecular weight or nucleic acid size markers must be included on all gel electrophoresis. Please add molecular weight with unit labels on the following panels: S2B please add unit labels

Scale bars have been added to the top panels of Fig. 3B, and the units added to Fig. S2B. All other figures have been checked.

5) Statistical analysis: Error bars on graphic representations of numerical data must be clearly described in the figure legend. The number of independent data points (n) represented in a graph must be indicated in the legend. Statistical methods should be explained in full in the materials and methods. For figures presenting pooled data the statistical measure should be defined in the figure legends.

Please indicate n/sample size/how many experiments the data are representative of: 1B, 2EF

N numbers have been added to Fig. 2E-F, Fig. S1, Sup table 1 and Sup table 2, with further details provided for Fig. 3C-D. Fig 1B illustrates a yes/no screening step designed to narrow down the most promising targets; this has been clarified in the text.

6) Materials and methods: Should be comprehensive and not simply reference a previous publication for details on how an experiment was performed. Please provide full descriptions in the text for readers who may not have access to referenced manuscripts.

- Please include database/vendor IDs for all plasmids, strains and cell lines (e.g., Addgene, ATCC, etc.) - even if described in other published work or gifted to you by other researchers. If they are not available, please detail the basic genetic features, even if previously described in other work/gifts.

- Please include the non-targeting control siRNA sequence, if made available to you from the manufacturer.

- Microscope image acquisition: The following information must be provided about the acquisition and processing of images:

- a. Make and model of microscope
- b. Type, magnification, and numerical aperture of the objective lenses
- c. Temperature
- d. imaging medium
- e. Fluorochromes

- f. Camera make and model
- g. Acquisition software
- h. Any software used for image processing subsequent to data acquisition. Please include details and types of operations involved (e.g., type of deconvolution, 3D reconstitutions, surface or volume rendering, gamma adjustments, etc.).

ATCC vendor cell line catalogue numbers, the sequence of the non-targeting control siRNA, and the specified details on the microscope used to visualise the cell line experiments have been added to the text.

7) References: There is no limit to the number of references cited in a manuscript. References should be cited parenthetically in the text by author and year of publication. A

- Please abbreviate the names of journals according to PubMed.
- Please note our formatting guidelines for preprints and please be sure to reformat the following ref -- ****both for the in-text citation and reference list citation****:

<http://jcb.rupress.org/reference-guidelines>

"Dobramysl, U., I.K. Jarsch, H. Shimo, Y. Inoue, B. Richier, J.R. Gadsby, J. Mason, A. Walrant, R. Butler, E. Hannezo, B.D. Simons, and J.L. Gallop. 2019. Constrained actin dynamics emerges from variable compositions of actin regulatory protein complexes. BioRxiv. 525725."

We have amended the preprint citation and reference, and changed the journal titles in the reference section to the abbreviated style.

8) A summary paragraph of all supplemental material should appear at the end of the Materials and methods section.

- Please include ~1 brief descriptive sentence per item.

We have added a paragraph summarising the online supplemental material at the end of the material and methods section.